# DUAL-AN: A HIERARCHICAL FRAMEWORK SYNERGIZES TIME AND FREQUENCY DOMAINS FOR NONSTATIONARY TIME SERIES FORECASTING

## ABSTRACT

To address the pervasive and challenging issue of non-stationarity in time series forecasting, recent research has primarily focused on time-domain normalization methods that separate non-stationary features using statistical indicators. The proposal of frequency adaptive normalization (FAN) offers a new perspective for separating non-stationary components in the frequency domain. However, existing methods remain confined to a single domain, lacking a synergistic integration of time and frequency domains. To bridge this gap, we introduce Dual-AN, a hierarchical framework that synergizes both time and frequency domains. After utilizing the Fourier transform approach to separate non-stationary factors, we propose a novel sliding window adaptive normalization (SWAN) method to eliminate the local non-stationarity in the residuals. Furthermore, we introduce the statistical prediction module (SPM) to forecast future statistics, which are used to de-normalize the outputs based on the statistics of each window. Dual-AN is a general framework that can be easily integrated into any forecasting model. We evaluate the improvement in forecasting performance of 3 different benchmark models on 8 widely-used datasets. The results show that Dual-AN demonstrates significant performance improvement, with the average prediction error MAE and MSE reduced by 15.92% and 20.72%. In comparison with other existing normalization methods, Dual-AN surpasses all existing methods and achieves state-of-the-art (SOTA) performance with an average prediction error reduction of 7.69%.

## 1 INTRODUCTION

Time series forecasting is of critical importance in numerous domains, including finance Li & Bastos (2020), medicine Bertozzi et al. (2020), energy Hong et al. (2020), transportation Ermagun & Levinson (2018), meteorology Murphy & Winkler (1984), and electricity Nti et al. (2020). However, traditional machine learning and deep learning approaches often struggle in forecasting tasks due to challenges such as distribution shift Kuang et al. (2020); Cao et al. (2022), which is a phenomenon inherent in non-stationary time series Hyndman & Athanasopoulos (2018). These dynamic properties pose significant obstacles to accurate prediction.

In recent years, the non-stationarity in time series has attracted growing attention. Since the introduction of the reversible normalization method in 2022 Kim et al. (2021), mainstream research has focused on exploiting time-domain statistics to mitigate non-stationary signals Fan et al. (2023); Liu et al. (2023c). More recently, frequency adaptive normalization (FAN) Ye et al. (2024) has opened a new direction by operating in the frequency domain. Instead of the normalization using time-domain statistics, FAN alleviates the impact of non-stationarity by selecting the top $K$ dominant components in the Fourier domain, thereby holistically handling composite non-stationary factors involving both trend and periodic components.

Nevertheless, using only the top $K$ dominant components in the Fourier domain to represent nonstationary information may be insufficient, as residuals often retain local non-stationarity Que et al. (2020), such as transient shocks in traffic data Zheng et al. (2011) or micro-trends in financial series Moon (2013). The residual learning strategy of FAN Ye et al. (2024) overlooks these fine-grained distribution shifts Deldari et al. (2021); Lai et al. (2021), violating the independent and identically

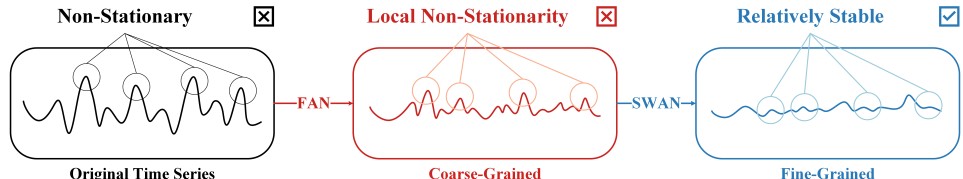

Figure 1: The comparison of our sliding window adaptive normalization (SWAN) and FAN. Our SWAN can eliminate the local non-stationarity in the time series and make it relatively stable at a fine granularity, while FAN cannot.

distributed assumption underlying many deep learning models. Simultaneously, most existing normalization techniques are confined to a single domain. While some end-to-end models have explored joint time-frequency representations Chen et al. (2023); Wu et al. (2022b), a dedicated, model-agnostic normalization framework that synergizes both domains is still lacking. To address this, the **Dual**-domain **A**daptive **N**ormalization (**Dual-AN**) is proposed, a hierarchical framework designed for universal integration with any forecasting backbone. In contrast to FAN Ye et al. (2024), we introduce a novel sliding window adaptive normalization (SWAN) method to eliminate the local non-stationarity in the residuals to better align with the input characteristics of the model, as illustrated in Figure 1. Additionally, we design a statistical prediction module (SPM) that forecasts future statistics using the statistics from each window to de-normalize the outputs, effectively combining fine-grained time-domain statistical features processing with coarse-grained frequency-domain decomposition. All code and data are available at https://anonymous.4open.science/r/Dual-AN. Our main contributions are summarized as follows:

- A novel, model-agnostic framework is presented that hierarchically addresses non-stationarity in both time and frequency domains. This approach overcomes the limitations of single-domain normalization methods, such as FAN's handling of local non-stationarity in residuals.

- We design a novel Sliding Window Adaptive Normalization (SWAN) method and a Statistical Prediction Module (SPM) that forecasts the future window-level statistics from frequency-domain residuals to de-normalize the outputs, enabling accurate reconstruction in the time domain.

- We conduct extensive experiments on 8 mainstream time series datasets. The results demonstrate that Dual-AN consistently improves performance across 3 backbone models, reducing average MAE and MSE by up to 15.92% and 20.72%, respectively. Moreover, it outperforms 4 existing normalization methods, including FAN, with an average MAE reduction of 7.69%, achieving the state-of-the-art (SOTA) performance and underscoring the superiority of our approach.

## 2 RELATED WORK

### 2.1 TIME SERIES FORECASTING

Time series forecasting is a critical task across numerous domains. Traditional statistical approaches like ARIMA Box & Jenkins (1968); Zhang (2003) rely on assumptions of stationarity and temporal dependency, which frequently do not hold in real-world scenarios. The advent of deep learning has significantly advanced the field, with architectures including CNNs LeCun et al. (2002); Lea et al. (2017); Liu et al. (2022a); Wang et al. (2023), RNNs/LSTMs Jordan (1997); Du et al. (2021); Lin et al. (2023); Hochreiter & Schmidhuber (1997), Transformers Vaswani et al. (2017); Zhou et al. (2021); Nie et al. (2022); Liu et al. (2023a); Wang et al. (2024b), and MLPs Rosenblatt (1958); Zeng et al. (2023); Das et al. (2023); Wang et al. (2024a); Murad et al. (2025) each contributing distinct strengths. CNN-based methods excel at capturing local patterns but struggle with long-range dependencies and non-stationary data Zheng et al. (2014). RNNs and LSTMs model sequential state transitions effectively but suffer from computational inefficiency and challenges in very long sequences Siami-Namini et al. (2019); Smyl (2020); Salinas et al. (2020); Hewamalage et al. (2021). Transformers leverage self-attention to capture global and cross-variable dependencies, yet face issues with computational complexity and sparse data Zhou et al. (2021). MLP-based models offer simplicity and scale well, but often fall short in modeling complex temporal relationships compared to recurrent or attention-based approaches Zhang et al.; Yi et al. (2023).

A crucial challenge across all architectures is handling non-stationary time series exhibiting distribution shifts Petropoulos et al. (2022) with the core of the modeling of time-varying statistical

properties, such as trend drift, seasonality, and shift points. Existing approaches include: (a) traditional stabilization via differencing, decomposition, or filtering Box & Jenkins (1968); Zhang (2003); Cleveland et al. (1990); Taylor & Letham (2018); Kalman (1960); (b) implicit modeling using RNNs Hochreiter & Schmidhuber (1997); Cho et al. (2014); Chung et al. (2014), enhanced attention Kitaev et al. (2020), or normalization techniques Ogasawara et al. (2010); Passalis et al. (2019); Deng et al. (2021); Kim et al. (2021); Fan et al. (2023); Liu et al. (2023c); Ye et al. (2024); (c) explicit decomposition architectures, which have recently become prominent—e.g., N-BEATS Oreshkin et al. (2019), ETSformer Woo et al. (2022b), Autoformer Wu et al. (2021), FEDformer Zhou et al. (2022), TimesNet Wu et al. (2022a), Pyraformer Liu et al. (2022b), Crossformer Zhang & Yan (2023), and Koopa Liu et al. (2023b); and (d) emerging trends such as frequency-domain analysis Xu et al. (2023); Yi et al. (2023), distributionally robust learning Woo et al. (2022a); Liu et al. (2022c); Zeng et al. (2023), change-point detection Adams & MacKay (2007); Xu & Zhu (2023), and improved benchmarks and evaluation Makridakis et al. (2018); Zhou et al. (2021); Challu et al. (2023). Despite these advances, modeling non-stationary time series remains an open and highly active research problem due to its practical significance and theoretical challenges.

## 2.2 NORMALIZATION METHODS AGAINST NON-STATIONARITY

Recent normalization methods have sought to mitigate non-stationarity, a primary obstacle in time series forecasting Ogasawara et al. (2010); Passalis et al. (2019); Deng et al. (2021); Kim et al. (2021); Fan et al. (2023); Liu et al. (2023c); Ye et al. (2024). These can be broadly categorized by their operating domain. Time-domain approaches, such as RevIN Kim et al. (2021)—a form of reversible instance normalization Ulyanov et al. (2016)—and Dish-TS Fan et al. (2023), utilize statistical moments to counteract distribution shifts. SAN Liu et al. (2023c) further refines this by employing adaptive local statistics. While effective against trends, these methods' reliance on statistics often proves insufficient for capturing complex seasonal variations. In contrast, FAN Ye et al. (2024) operates in the frequency domain, isolating dominant components to jointly model trend and seasonality. Despite these advances, a clear dichotomy persists: methods operate largely in either the time domain Kim et al. (2021); Fan et al. (2023); Liu et al. (2023c) or the frequency domain Ye et al. (2024). While another line of research develops end-to-end architectures that jointly process time-frequency information Chen et al. (2023); Wu et al. (2022b), their monolithic, architecturally-specific nature prevents their use as universal modules. This context reveals a critical gap: the lack of a model-agnostic framework that synergizes both domains. The proposed Dual-AN is conceptualized to fill this void. It performs a coarse-grained frequency decomposition followed by a fine-grained, adaptive time-domain normalization on the residual series, offering a versatile tool to enhance any existing forecasting backbone.

## 2.3 MODEL-AGNOSTIC PLUG-IN METHODS

Recent works also design model-agnostic plug-in modules that can be seamlessly attached to diverse time series forecasting (TSF) backbones. DDN Dai et al. (2024) performs dual-domain dynamic normalization via sliding-window statistics in time and frequency domains, while BSA Kang et al. (2024) introduces a batched spectral attention block to capture long-range dependencies in the spectral space. SCAM Yang et al. (2025) and HCAN Sun et al. (2025) instead focus on the supervision signal: SCAM corrects noisy labels by self-generated pseudo labels with adaptive masks, and HCAN adds a hierarchical classification auxiliary head to shape multi-scale representations. TAFAS Kim et al. (2025) tackles test-time distribution shift by adapting pre-trained forecasters online on unlabeled target streams. Our Dual-AN framework is complementary to these approaches: it acts as a lightweight, plug-and-play module that explicitly synergizes coarse frequency-domain decomposition with fine-grained time-domain normalization and future-statistics prediction, aiming to stabilize non-stationarity at the data level and thus providing a generic improvement that can, in principle, be combined with the above plug-ins.

## 3 DUAL-AN

The proposed Dual-AN method operates via a hierarchical, dual-domain process to address non-stationarity, as illustrated in Figure 2. Following an initial frequency-domain decomposition that

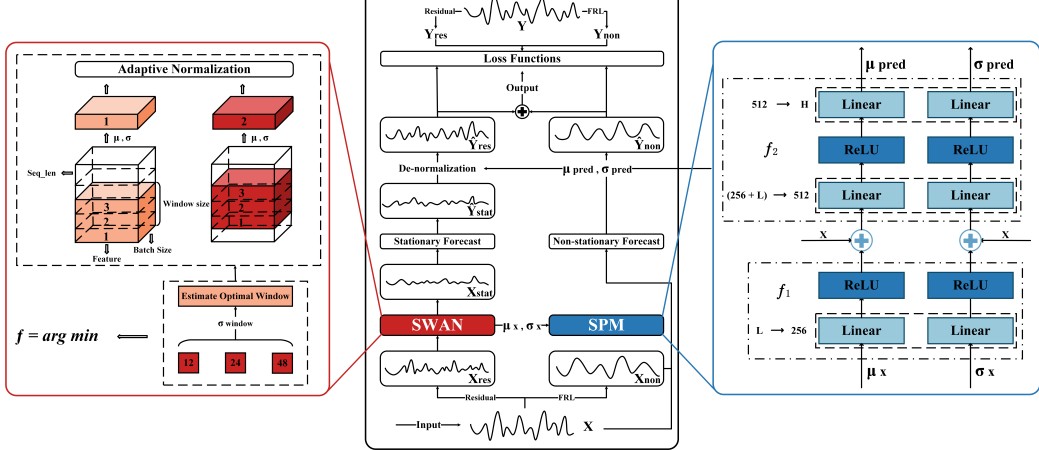

Figure 2: The overall architecture of Dual-AN, highlighting its two core modules: Sliding Window Adaptive Normalization (SWAN) and Statistical Prediction Module (SPM). The process begins with Frequency Residual Learning (FRL, see Appendix A.5) to obtain residuals. SWAN then normalizes these residuals to address local non-stationarity, and SPM predicts future statistics for the final de-normalization step. Detailed algorithms are provided in Appendix C.1 and C.2.

isolates coarse-grained non-stationary signals, two core modules are introduced: the Sliding Window Adaptive Normalization (SWAN) and the Statistical Prediction Module (SPM). SWAN targets the remaining local non-stationarity within the time-domain residuals, while SPM forecasts future window-level statistics to enable precise, adaptive reconstruction of the final prediction. The stationary component is forecasted by a backbone model, whereas the non-stationary component is handled by a dedicated MLP network.

## 3.1 SLIDING WINDOW ADAPTIVE NORMALIZATION (SWAN)

Since there may still be local non-stationarity in the residuals after frequency domain separation, we design a normalization method in the time domain that pays more attention to the local characteristics of the data, namely, sliding window adaptive normalization (SWAN), which uses the dynamic selection and adaptive normalization of the sliding window to standardize the time series data. For multivariate time series, the SWAN process is applied independently to each variable (channel-wise). This approach ensures that the unique statistical properties and scales of each channel are preserved, preventing cross-channel distortion during normalization.

### 3.1.1 DYNAMIC OPTIMAL WINDOW SIZE SELECTION

In order to determine the size of the dynamic window, we evaluate the local standard deviation of different window sizes to dynamically select the optimal size. For each defined valid candidate window size, we fill the inputs to ensure that it can be applied to every position of the data.

Afterwards, for each window size in the set of the candidate window sizes $\phi = \{12, 24, 48\}$, we compute the local standard deviation. Specifically, for each time step $t$, calculate the standard deviation $\sigma_{window}$ of the data in the window at that time step:

$$\sigma_{\text{window}}(\text{i}) = \text{std}(\text{x}[\text{i} : \text{i} + \text{window}, :]), \tag{1}$$

where $i = 1, 2, \ldots, L$, then we compute the standard deviation of the local standard deviations over all time steps for that window:

$$\text{SD}_{\text{window}} = \text{std}(\sigma_{\text{window}}(\text{i})), \tag{2}$$

where $i = 1, 2, \ldots, L$ and $L$ is the length of the sequence. A lower $SD$ value indicates that the local volatility of the series is more consistent at that specific window scale. Selecting a window size that yields such statistical homogeneity is hypothesized to produce a more uniformly normalized sequence, better satisfying the stationarity assumption required by the downstream forecasting

model. While this criterion is heuristic, its empirical effectiveness is validated in Section 4.5.3. A deeper discussion on this selection principle is provided in Appendix F. Finally, the window size with the lowest $SD$ value is selected as the optimal window:

$$\mathrm{W_{Best\_window}} = \arg \min_\phi \mathrm{SD_{window}} \tag{3}$$

### 3.1.2 SLIDING WINDOW ADAPTIVE NORMALIZATION

After selecting the optimal sliding window size, we use the adaptive normalization method to normalize the inputs according to the selected window size.

First, we pad the data with a padding size of half the window. For each time step $i$, we use the optimal window size $W_{optimal}$ to calculate the mean and standard deviation of the data in the window:

$$\mu_{window}(i) = \frac{1}{W} \sum_{j=i}^{W} X_j, \tag{4}$$

$$\sigma_{window}(i) = \sqrt{\frac{1}{W} \sum_{j=i}^{W} (X_j - \mu_{window}(i))^2} \tag{5}$$

Afterwards, for each time step $i$, the selected dynamic window size slides across the input sequence, and the central value at each time step is normalized using statistics derived from its own local temporal neighborhood. Specifically, the value at the center is standardized by subtracting the window mean and dividing by the window standard deviation:

$$\mathrm{X_{stat}(i)} = \frac{\mathrm{X(i)} - \mu_{window}(i)}{\sigma_{window}(i) + \varepsilon} \tag{6}$$

where $\varepsilon = 1e - 5$ is a small constant to prevent the standard deviation from being zero.

### 3.2 STATISTICAL PREDICTION MODULE (SPM)

In Section 3.1.2, we retain the mean and standard deviation of each window as statistical indicators in the time domain. In order to reflect the statistical characteristics of the forecasting results, a statistical prediction module (SPM) is designed to forecast the mean and standard deviation for future windows. An MLP architecture is selected for the SPM due to its balance of expressive power as a universal function approximator and computational efficiency. This design is sufficient for predicting the smoother statistical moment sequences while avoiding the substantial overhead of more complex sequential models (e.g., RNNs). The rationale for this design choice is further detailed in Appendix G. The module is formalized as:

$$\widehat{\mu}_{window} = f_2(Concat(f_1(\mu_{window}), X)), \tag{7}$$

$$\widehat{\sigma}_{window} = f_2(Concat(f_1(\sigma_{window}), X)) \tag{8}$$

where $f_1$ and $f_2$ represent 2 different multi-layer perceptron (MLP) networks as depicted in Figure 2 and Appendix C.2. Afterwards, the outputs are de-normalized using the predicted statistical indicators to obtain the predicted stationary component results $\hat{Y}_{res}$:

$$\hat{\mathrm{Y}}_{\mathrm{res}} = \hat{\mathrm{Y}}_{\mathrm{stat}} \cdot \widehat{\sigma}_{\mathrm{window}} + \widehat{\mu}_{\mathrm{window}} \tag{9}$$

where $\hat{Y}_{stat}$ represents the result predicted by the backbone network with the input $X_{stat}$. Finally, this part will be added to the non-stationary part $\hat{Y}_{non}$ predicted above to get the final forecasting results $\hat{Y}$.

## 3.3 Loss Functions

The model is optimized via a dual-component loss function that separately supervises the non-stationary and stationary predictions. This structure acts as a powerful regularization mechanism, guiding the model toward a more meaningful decomposition by ensuring both components are independently accurate. An ablation study presented in Appendix G confirms that this dual-objective approach yields superior performance compared to a single loss on the final output. The overall loss function is defined as:

$$\phi, \theta = \arg \min_{\phi, \theta} \sum_i \left( \mathcal{L}_{nonstat} + \mathcal{L}_{stat} \right), \tag{10}$$

$$\mathcal{L}_{nonstat} = \left( \mathcal{L}_\phi^{nonstat}(\mathbf{Y}_{non}(i), \hat{\mathbf{Y}}_{non}(i)) \right), \tag{11}$$

$$\mathcal{L}_{stat} = \left( \mathcal{L}_{\theta,\phi}^{stat}(\mathbf{Y}_{stat}(i), \hat{\mathbf{Y}}_{stat}(i)) \right). \tag{12}$$

where $\phi$ and $\theta$ denote the learnable parameters of the forecasting model, and both loss functions are computed using the mean square error (MSE):

$$\mathcal{L}_{MSE} = \frac{\sum_{i=1}^n (y_i - \hat{y}_i)^2}{n} \tag{13}$$

where $n$ is the number of samples, $y_i$ is the ground truth of the $i^{th}$ sample, and $\hat{y}_i$ is the corresponding predicted value.

## 4 Experiments

This study conducts extensive experiments on the Dual-AN method using 8 widely used datasets in the field of time series forecasting to demonstrate its excellent performance.

### 4.1 Experimental Design

In this section, we introduce the datasets used in the experiments and the experimental settings to ensure the reproducibility of this paper.

**Datasets.** We use 8 of the most popular open source datasets in the time series field, including (1)ETTh1, (2)ETTh2, (3)ETTm1, (4)ETTm2, (5)Electricity, (6)Exchange Rate, (7)Traffic, and (8)Weather. In the preprocessing stage, we followed the practice in the FAN Ye et al. (2024) method and applied z-score normalization Goodfellow et al. (2016) to all datasets. The training set, validation set, and test set split ratio were set to 7:2:1, while retaining the setting of its hyperparameter $K$. For detailed properties and characteristics of the datasets, please refer to Appendix B.1.

**Experimental Setup.** To cover both short-term and long-term forecasts, we set the forecast length $H \in \{96, 168, 336, 720\}$, and all datasets use a fixed input length $L = 96$. We use the mean absolute error (MAE) and the mean square error (MSE) as metrics to evaluate the performance of the model, which are defined in Appendix B.2. Since Dual-AN is a universal plug-in, it can be applied to any backbone model for forecasting. To verify its effectiveness, we use 3 of the most common time series forecasting models as benchmark models: (1) DLinear Zeng et al. (2023), based on the multi-layer perceptron (MLP) network; (2) Informer Zhou et al. (2021), based on Transformer; (3) SCINet Liu et al. (2022a), based on the convolutional neural network (CNN). For the implementation details, all experiments in this paper are implemented by PyTorch Paszke et al. (2019) and tested in 5 rounds using fixed random seeds $\{1, 2, 3, 4, 5\}$ on NVIDIA RTX 4090 GPU (24GB).

### 4.2 Main Experimental Results of Dual-AN

We show the MAE and MSE metrics of the baseline model and Dual-AN on 5 datasets in Table 1. Please see Table 10 in the Appendix D.2 for full results of all 8 datasets.

The empirical results, summarized in Table 1, demonstrate that integrating Dual-AN yields substantial and consistent performance gains across all three backbone models and eight benchmark

Table 1: Main experimental results with and without Dual-AN. The best results are highlighted in **bold**.

| Models | | DLinear | | +Dual-AN | | Informer | | +Dual-AN | | SCINet | | +Dual-AN | |
|---|---|---|---|---|---|---|---|---|---|---|---|---|---|
| Metrics | | MAE | MSE | MAE | MSE | MAE | MSE | MAE | MSE | MAE | MSE | MAE | MSE |
| ETTh2 | 96 | 0.237 | **0.110** | **0.236** | **0.110** | 0.298 | 0.160 | **0.238** | **0.111** | 0.264 | 0.128 | **0.237** | **0.112** |
| | 168 | 0.254 | 0.127 | **0.250** | **0.125** | 0.331 | 0.191 | **0.252** | **0.127** | 0.292 | 0.156 | **0.249** | **0.125** |
| | 336 | 0.271 | 0.138 | **0.264** | **0.138** | 0.347 | 0.208 | **0.276** | **0.147** | 0.305 | 0.167 | **0.262** | **0.137** |
| | 720 | 0.316 | 0.179 | **0.280** | **0.157** | 0.413 | 0.291 | **0.337** | **0.208** | 0.339 | 0.201 | **0.284** | **0.156** |
| ETTm2 | 96 | 0.203 | 0.080 | **0.199** | **0.078** | 0.226 | 0.091 | **0.199** | **0.079** | 0.206 | 0.079 | **0.199** | **0.078** |
| | 168 | 0.220 | **0.093** | 0.219 | 0.093 | 0.251 | 0.112 | **0.220** | **0.093** | 0.226 | 0.094 | **0.219** | **0.093** |
| | 336 | 0.245 | 0.114 | **0.242** | **0.113** | 0.283 | 0.140 | **0.245** | **0.114** | 0.262 | 0.122 | **0.242** | **0.113** |
| | 720 | 0.270 | 0.142 | **0.264** | **0.139** | 0.347 | 0.212 | **0.277** | **0.147** | 0.297 | 0.153 | **0.264** | **0.139** |
| Electricity | 96 | 0.277 | 0.195 | **0.265** | **0.181** | 0.376 | 0.277 | **0.244** | **0.148** | 0.296 | 0.188 | **0.254** | **0.159** |
| | 168 | 0.272 | 0.183 | **0.265** | **0.176** | 0.371 | 0.269 | **0.254** | **0.159** | 0.306 | 0.196 | **0.256** | **0.160** |
| | 336 | 0.294 | 0.197 | **0.285** | **0.190** | 0.377 | 0.273 | **0.270** | **0.166** | 0.330 | 0.214 | **0.272** | **0.169** |
| | 720 | 0.333 | 0.233 | **0.320** | **0.223** | 0.401 | 0.311 | **0.302** | **0.191** | 0.352 | 0.240 | **0.303** | **0.194** |
| Traffic | 96 | 0.387 | 0.504 | **0.334** | **0.403** | 0.350 | 0.428 | **0.323** | **0.386** | 0.399 | 0.471 | **0.325** | **0.393** |
| | 168 | 0.588 | 0.804 | **0.333** | **0.413** | 0.366 | 0.457 | **0.320** | **0.393** | 0.377 | 0.443 | **0.328** | **0.408** |
| | 336 | 0.380 | 0.504 | **0.345** | **0.436** | 0.414 | 0.555 | **0.336** | **0.425** | 0.384 | 0.459 | **0.345** | **0.436** |
| | 720 | 0.407 | 0.532 | **0.368** | **0.469** | 0.656 | 1.002 | **0.356** | **0.448** | 0.401 | 0.490 | **0.368** | **0.469** |
| Weather | 96 | 0.249 | **0.180** | **0.220** | 0.181 | 0.299 | 0.221 | **0.210** | **0.172** | 0.265 | 0.199 | **0.211** | **0.170** |
| | 168 | 0.284 | 0.237 | **0.259** | **0.218** | 0.363 | 0.320 | **0.250** | **0.211** | 0.305 | 0.245 | **0.252** | **0.209** |
| | 336 | 0.344 | 0.304 | **0.298** | **0.278** | 0.439 | 0.437 | **0.301** | **0.270** | 0.341 | 0.310 | **0.293** | **0.271** |
| | 720 | 0.380 | 0.358 | **0.346** | **0.343** | 0.496 | 0.524 | **0.366** | **0.349** | 0.383 | 0.371 | **0.331** | **0.329** |

datasets. The framework reduces the average prediction error by up to 15.92% in MAE and 20.72% in MSE, confirming its effectiveness in mitigating the adverse effects of non-stationarity.

A key observation is that the performance improvement is particularly pronounced in long-term forecasting scenarios. For instance, when applied to the Informer backbone, the error reduction escalates with the prediction horizon, underscoring the framework's capability to preserve long-range temporal dependencies. This enhanced long-term performance is attributed to a virtuous cycle created by Dual-AN: by providing a more stable, stationary input, it enables the backbone model to learn more generalizable temporal patterns, which in turn prevents the error accumulation that typically plagues long-horizon forecasts in non-stationary series. These findings highlight the efficacy of the proposed hierarchical normalization approach, especially for challenging long-horizon forecasting tasks.

## 4.3 COMPARATIVE EXPERIMENTS WITH EXISTING NORMALIZATION METHODS

To benchmark Dual-AN against its direct peers, we compare it with leading model-agnostic normalization frameworks designed for non-stationarity: FAN Ye et al. (2024), SAN Liu et al. (2023c), Dish-TS Fan et al. (2023), and RevIN Kim et al. (2021). Table 2 summarizes the resulting MAE scores across all settings.

Table 2: Averaged MAE performance compared with other normalization methods. The best performance is highlighted in red and the second best performance is underlined. Please see Table 11 in the Appendix D.2 for full results.

| Models | DLinear | | | | | Informer | | | | | SCINet | | | | |
|---|---|---|---|---|---|---|---|---|---|---|---|---|---|---|---|
| Methods | Dual-AN | FAN | SAN | Dish-TS | RevIN | Dual-AN | FAN | SAN | Dish-TS | RevIN | Dual-AN | FAN | SAN | Dish-TS | RevIN |
| ETTh1 | 0.484 | 0.484 | 0.495 | 0.496 | 0.498 | 0.485 | 0.502 | 0.582 | 0.640 | 0.616 | 0.487 | 0.485 | 0.493 | 0.514 | 0.496 |
| ETTh2 | 0.257 | 0.257 | 0.260 | 0.262 | 0.268 | 0.276 | 0.301 | 0.324 | 0.376 | 0.329 | 0.258 | 0.262 | 0.264 | 0.291 | 0.271 |
| ETTm1 | 0.439 | 0.440 | 0.439 | 0.447 | 0.457 | 0.444 | 0.444 | 0.470 | 0.524 | 0.509 | 0.438 | 0.440 | 0.441 | 0.463 | 0.476 |
| ETTm2 | 0.231 | 0.231 | 0.231 | 0.237 | 0.238 | 0.235 | 0.237 | 0.241 | 0.284 | 0.259 | 0.231 | 0.230 | 0.229 | 0.249 | 0.236 |
| Electricity | 0.284 | 0.286 | 0.300 | 0.297 | 0.290 | 0.267 | 0.269 | 0.303 | 0.329 | 0.295 | 0.271 | 0.277 | 0.284 | 0.310 | 0.267 |
| Exchange | 0.268 | 0.272 | 0.287 | 0.360 | 0.305 | 0.278 | 0.295 | 0.353 | 0.485 | 0.349 | 0.275 | 0.282 | 0.290 | 0.386 | 0.300 |
| Traffic | 0.345 | 0.347 | 0.414 | 0.451 | 0.484 | 0.334 | 0.341 | 0.407 | 0.371 | 0.575 | 0.342 | 0.355 | 0.359 | 0.402 | 0.369 |
| Weather | 0.281 | 0.278 | 0.289 | 0.319 | 0.269 | 0.282 | 0.287 | 0.292 | 0.346 | 0.277 | 0.272 | 0.277 | 0.285 | 0.293 | 0.268 |
| Count ($1^{st}$) | 7 | 3 | 2 | 0 | 1 | 7 | 1 | 0 | 0 | 1 | 4 | 1 | 1 | 0 | 2 |

Dual-AN demonstrates superior performance across most datasets, with the notable exception of the Weather dataset. Here, RevIN Kim et al. (2021) excels, an insightful finding we attribute to this dataset's very weak trend and seasonality (see Appendix B.1). In such scenarios, the benefits of frequency decomposition are marginal, making simpler, moment-based normalization sufficient. This highlights a key characteristic: Dual-AN's strength is most pronounced on series with complex,

multi-scale non-stationarity, a common trait in real-world applications. Additionally, we observe a slightly diminished gain on the SCINet Liu et al. (2022a) backbone, likely due to an overlap between its sub-sequence decomposition and SWAN's focus on local patterns.

As shown in Table 2, Dual-AN reduces the average MAE by 1.50% (vs. FAN), 6.30% (vs. SAN), 14.17% (vs. Dish-TS), and 8.79% (vs. RevIN). Excluding the Weather dataset, these improvements are even more significant, reaffirming the strong and consistent performance of our framework.

To further illustrate the superiority of the proposed Dual-AN method, we compare it with state-of-the-art plug-in methods Kang et al. (2024); Dai et al. (2024); Sun et al. (2025); Yang et al. (2025); Kim et al. (2025) in Table 3 with the average MAE/MSE reduction rate of 15.78%/37.68% (vs. DDN), 15.60%/36.85% (vs. HCAN), 17.30%/35.12% (vs. BSA), 12.17%/35.36% (vs. SCAM), and 17.81%/33.55% (vs. TAFAS).

Table 3: Full results of the comparison of Dual-AN with other state-of-the-art plug-in methods on ETTh1, ETTh2, ETTm2, Exchange Rate and Traffic datasets using iTransformer as the backbone. The best performance is highlighted in red and the second best performance is underlined.

| Datasets | Horizons | Metrics | iTransformer | +Dual-AN | +DDN | +HCAN | +BSA | +SCAM | +TAFAS |
|---|---|---|---|---|---|---|---|---|---|
| ETTh1 | 96 | MAE | 0.444 | 0.426 | 0.399 | 0.402 | 0.443 | 0.401 | 0.443 |
| | | MSE | 0.378 | 0.362 | 0.388 | 0.379 | 0.428 | 0.373 | 0.438 |
| | 192 | MAE | 0.489 | 0.452 | 0.434 | 0.427 | 0.481 | 0.436 | 0.489 |
| | | MSE | 0.431 | 0.395 | 0.446 | 0.432 | 0.481 | 0.432 | 0.492 |
| | 336 | MAE | 0.533 | 0.486 | 0.462 | 0.454 | 0.521 | 0.455 | 0.532 |
| | | MSE | 0.511 | 0.441 | 0.496 | 0.489 | 0.538 | 0.466 | 0.554 |
| | 720 | MAE | 0.64 | 0.569 | 0.499 | 0.474 | 0.62 | 0.466 | 0.627 |
| | | MSE | 0.669 | 0.574 | 0.527 | 0.504 | 0.698 | 0.455 | 0.704 |
| ETTh2 | 96 | MAE | 0.255 | 0.237 | 0.345 | 0.343 | 0.324 | 0.342 | 0.329 |
| | | MSE | 0.122 | 0.111 | 0.297 | 0.282 | 0.235 | 0.293 | 0.239 |
| | 192 | MAE | 0.282 | 0.252 | 0.397 | 0.381 | 0.362 | 0.393 | 0.362 |
| | | MSE | 0.148 | 0.128 | 0.382 | 0.373 | 0.29 | 0.373 | 0.287 |
| | 336 | MAE | 0.3 | 0.264 | 0.431 | 0.426 | 0.388 | 0.429 | 0.386 |
| | | MSE | 0.167 | 0.139 | 0.419 | 0.42 | 0.327 | 0.417 | 0.326 |
| | 720 | MAE | 0.362 | 0.279 | 0.446 | 0.435 | 0.439 | 0.442 | 0.425 |
| | | MSE | 0.482 | 0.155 | 0.426 | 0.423 | 0.414 | 0.424 | 0.393 |
| ETTm2 | 96 | MAE | 0.203 | 0.199 | 0.265 | 0.264 | 0.259 | 0.264 | 0.263 |
| | | MSE | 0.078 | 0.078 | 0.181 | 0.183 | 0.153 | 0.179 | 0.157 |
| | 192 | MAE | 0.239 | 0.222 | 0.303 | 0.312 | 0.29 | 0.302 | 0.292 |
| | | MSE | 0.103 | 0.095 | 0.246 | 0.242 | 0.189 | 0.241 | 0.192 |
| | 336 | MAE | 0.247 | 0.243 | 0.342 | 0.355 | 0.321 | 0.343 | 0.324 |
| | | MSE | 0.114 | 0.114 | 0.306 | 0.306 | 0.23 | 0.305 | 0.235 |
| | 720 | MAE | 0.277 | 0.264 | 0.397 | 0.401 | 0.369 | 0.4 | 0.366 |
| | | MSE | 0.144 | 0.139 | 0.406 | 0.41 | 0.304 | 0.406 | 0.301 |
| Exchange | 96 | MAE | 0.212 | 0.164 | 0.202 | 0.204 | 0.211 | - | 0.208 |
| | | MSE | 0.081 | 0.051 | 0.084 | 0.084 | 0.09 | - | 0.084 |
| | 192 | MAE | 0.331 | 0.238 | 0.297 | 0.302 | 0.307 | - | 0.293 |
| | | MSE | 0.184 | 0.102 | 0.175 | 0.179 | 0.185 | - | 0.165 |
| | 336 | MAE | 0.504 | 0.324 | 0.41 | 0.415 | 0.43 | - | 0.389 |
| | | MSE | 0.398 | 0.178 | 0.321 | 0.322 | 0.346 | - | 0.28 |
| | 720 | MAE | 0.671 | 0.465 | 0.7 | 0.761 | 0.7 | - | 0.665 |
| | | MSE | 0.747 | 0.331 | 0.859 | 0.995 | 0.861 | - | 0.773 |
| Traffic | 96 | MAE | 0.3 | 0.297 | 0.271 | 0.262 | 0.273 | 0.247 | 0.289 |
| | | MSE | 0.338 | 0.349 | 0.425 | 0.383 | 0.393 | 0.374 | 0.42 |
| | 192 | MAE | 0.313 | 0.294 | 0.28 | 0.273 | 0.281 | 0.259 | 0.296 |
| | | MSE | 0.362 | 0.353 | 0.446 | 0.411 | 0.417 | 0.399 | 0.441 |
| | 336 | MAE | 0.319 | 0.3 | 0.291 | 0.279 | 0.29 | 0.269 | 0.305 |
| | | MSE | 0.375 | 0.364 | 0.459 | 0.42 | 0.433 | 0.419 | 0.458 |
| | 720 | MAE | 0.338 | 0.326 | 0.311 | 0.296 | 0.31 | 0.291 | - |
| | | MSE | 0.403 | 0.395 | 0.5 | 0.449 | 0.47 | 0.451 | - |
| Count(1st) | | | - | 31 | 1 | 2 | 0 | 6 | 0 |

## 4.4 ABLATION STUDY

This section evaluates the effectiveness of the two core components of the Dual-AN method, SWAN and SPM. We compare two ablation variants against Dual: "w/o SWAN" removes the sliding window adaptive normalization (SWAN) module, rendering the statistical prediction module (SPM) inactive due to the absence of statistical indicators; "w/o SPM" removes the statistical prediction module (SPM), instead using the original statistics of the inputs directly as the statistical indicators for de-normalization. Experiments are conducted on ETTh1 and Electricity datasets using Informer and SCINet as backbones, respectively, with MAE and MSE results summarized in Table 4.

Table 4: MAE and MSE indicators of ablation studies. The best results are highlighted in **bold**.

| Models | Metrics | ETTh1 | | | | Electricity | | | |
|---|---|---|---|---|---|---|---|---|---|
| | | 96 | 168 | 336 | 720 | 96 | 168 | 336 | 720 |
| Dual-AN | MAE | **0.431** | **0.446** | **0.493** | **0.579** | **0.254** | **0.256** | **0.272** | **0.303** |
| | MSE | **0.365** | **0.386** | **0.452** | **0.589** | **0.159** | **0.160** | **0.169** | **0.194** |
| w/o SWAN | MAE | 0.434 | 0.465 | 0.507 | 0.602 | 0.258 | 0.258 | 0.278 | 0.312 |
| | MSE | 0.367 | 0.407 | 0.467 | 0.617 | 0.165 | 0.163 | 0.175 | 0.204 |
| w/o SPM | MAE | 0.441 | 0.472 | 0.513 | 0.604 | 0.264 | 0.262 | 0.280 | 0.305 |
| | MSE | 0.381 | 0.418 | 0.473 | 0.617 | 0.170 | 0.166 | 0.177 | 0.199 |

The results demonstrate that Dual-AN consistently achieves the best performance across all ablation variants, confirming the importance of both the SWAN and SPM modules proposed in this study. The ablation variant w/o SWAN ranks second, slightly outperforming the variant w/o SPM. This performance gap stems from the fact that statistical indicators derived directly from the original time series fail to accurately capture future trends, leading to suboptimal de-normalization and thus degrading forecasting performance.

## 4.5 MODEL ANALYSIS

In this section, we discuss and analyze the parameters of the model, including the lookback length, horizon length, and the hyperparameter sliding window size.

### 4.5.1 LOOKBACK AND HORIZON ANALYSIS

We analyze the effects of the lookback and horizon lengths on the forecasting performance of Dual-AN on the Exchange Rate dataset on Informer and SCINet backbones, respectively, compared with FAN, which is the current state-of-the-art (SOTA) normalization method. We illustrate the experimental results in Figure 3, and the lookback and horizon lengths are set to $L \in \{48, 72, 96, 120, 144, 168\}$ and $H \in \{270, 336, 420, 540, 600, 720\}$, while keeping $H = 96$ and $L = 96$ respectively.

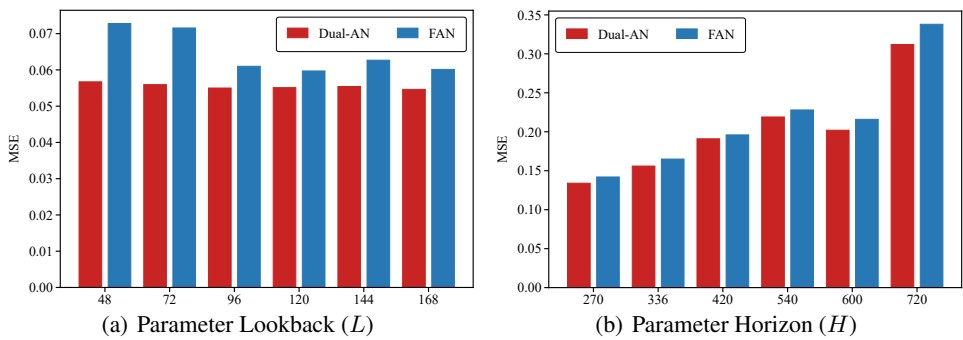

(a) Parameter Lookback ($L$)    (b) Parameter Horizon ($H$)

Figure 3: The MSE indicator of Dual-AN and FAN under different lookback and horizon settings. Please see Table 12 in the Appendix D.2 for full results.

As shown in Figure 3, Dual-AN consistently outperforms FAN across all lookback and horizon lengths. Notably, as the prediction horizon increases from 270 steps to 720 steps, the improvement gains of Dual-AN over FAN gradually increases with the reduction rate of MSE rising from 5.63% to 7.69%, which demonstrates the significant advantages of Dual-AN with the characteristics of coordinating time and frequency domains, especially in long-term time series forecasting.

### 4.5.2 CANDIDATE WINDOW SIZE

In the sliding window adaptive normalization (SWAN) module of our Dual-AN method, the size of the sliding window is a critical hyperparameter. In order to illustrate the rigor of the experiments

in this paper, we rigorously evaluate the impact of different window sizes on our method. We conduct experiments on the ETTm2 dataset using the DLinear backbone for the hyperparameter sliding window size. Since the lookback length is set to $L = 96$, we test 5 reasonable candidate window sizes $W_{exp} \in \{6, 12, 24, 48, 72\}$, and record the MAE and MSE indicators in Table 5.

Table 5: MAE and MSE indicators of the different window sizes. The best results are highlighted in **bold**.

| Window Size | 6 | | 12 | | 24 | | 48 | | 72 | |
|---|---|---|---|---|---|---|---|---|---|---|
| Metrics | MAE | MSE | MAE | MSE | MAE | MSE | MAE | MSE | MAE | MSE |
| 96 | 0.19876 | 0.07819 | **0.19871** | 0.07813 | 0.19887 | 0.07812 | 0.19884(4) | **0.07805** | 0.19883(7) | 0.07822 |
| 168 | 0.21911 | 0.09329(0) | **0.21893** | 0.09329(1) | 0.21896 | **0.09325** | 0.21983 | 0.09367 | 0.21899 | 0.09327 |
| 336 | 0.24262 | 0.11430 | 0.24252 | 0.11431 | 0.24286 | 0.11447 | **0.24153** | **0.11310** | 0.24275 | 0.11436 |
| 720 | 0.26464 | 0.13932 | 0.26448 | 0.13939 | **0.26446** | 0.13940 | 0.26450 | **0.13929** | 0.26455 | 0.13934 |
| Count ($1^{st}$) | 0 | 0 | **2** | 0 | **1** | **1** | **1** | **3** | 0 | 0 |

Experimental results show optimal performance is achieved with window sizes $W \in 12, 24, 48$, a range adopted for the main experiments in Section 3.1. This range effectively balances the trade-off between capturing sufficient context and preserving local temporal patterns. Moreover, the low performance variance across these optimal window sizes highlights Dual-AN's robustness to this hyperparameter choice. For more discussion of the window size selection, please refer to Appendix F.

### 4.5.3 VISUALIZATIONS

Figure 4 visualizes the performance gains of Dual-AN over the Informer backbone. The baseline model frequently fails to capture local extrema, a shortcoming that Dual-AN effectively addresses. This corrective capability is especially pronounced in long-horizon forecasting ($H = 720$), where the framework's advantage is most evident. Further visual comparisons are available in Appendix E.

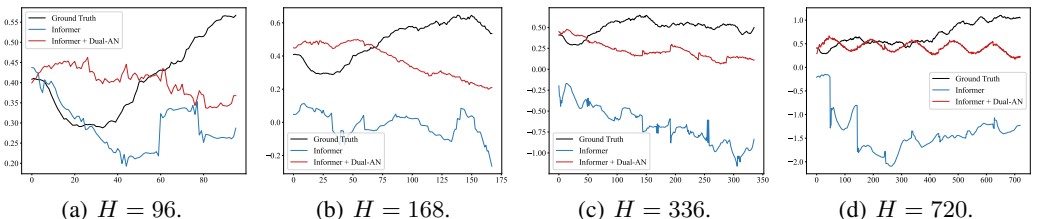

(a) $H = 96$.     (b) $H = 168$.     (c) $H = 336$.     (d) $H = 720$.

Figure 4: The visual forecasting results of backbone (Informer) and Dual-AN on the Weather dataset across 4 different prediction lengths.

## 5 CONCLUSION

In this paper, we propose Dual-AN, a general framework that synergizes time and frequency domains to address non-stationarity in time series forecasting. Its core components, the sliding window adaptive normalization (SWAN) and the statistical prediction module (SPM), respectively eliminate local residual non-stationarity and predict future statistics for de-normalization. Extensive experiments demonstrate that Dual-AN consistently enhances three backbone models, achieving state-of-the-art (SOTA) performance over existing normalization methods. Its feasibility as a lightweight, efficient plug-in is confirmed by a formal complexity analysis (Appendix I). For reproducibility, all source code and data are detailed in Section 6. Limitations and potential future directions are discussed in Appendix K.

## 6 REPRODUCIBILITY STATEMENT

In full compliance with double-blind review guidelines, we have taken extensive measures to ensure the reproducibility of our work. All source code and data from this study have been uploaded to the supplementary materials and have been made publicly available in an anonymous repository: https://anonymous.4open.science/r/Dual-AN. We have also included instructions for running the code and reproducing the results in the README file. Furthermore, all of these will be publicly released on GitHub immediately after the review process is completed to ensure reproducibility and facilitate future research in the broader field of time series forecasting.

## 7 ETHICS STATEMENT

We affirm that this work adheres to the ICLR Code of Ethics. All datasets used in this study are publicly available and widely accepted in the time series forecasting community. We conducted no human subject experiments, and all data are anonymized and aggregated, posing no privacy or security risks. Our proposed method, Dual-AN, is a general forecasting framework and does not target sensitive or high-risk applications. However, we acknowledge that time series forecasting models can potentially be misapplied in domains such as surveillance, financial manipulation, or discriminatory decision-making. We strongly discourage any such misuse. The research was conducted with integrity, and we declare no conflicts of interest. All authors have read and complied with the ICLR Code of Ethics.

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

# A  PRELIMINARIES

In this section, we introduce the basics of this study from the aspects of multivariate time series forecasting, trend variation and seasonality variation, Fast Fourier Transform (FFT), Discrete Fourier Transform (DFT) and Inverse Discrete Fourier Transform (IDFT), and the frequency adaptive normalization (FAN) method Ye et al. (2024).

## A.1  MULTIVARIATE TIME SERIES FORECASTING

As for the multivariate time series forecasting, we denote multiple time series as $\mathbf{X}_t \in \mathbb{R}^{N \times L}$, where $N$ is the number of variables of the time series and each time series has a lookback length of $L$ at timestamp $t$. Then we use the forecasting model $\mathcal{F}$ to predict the future time series $(\hat{\mathbf{X}}_{t+1}, \hat{\mathbf{X}}_{t+2}, ..., \hat{\mathbf{X}}_{t+H})$ based on the historical time series $(\mathbf{X}_{t-L+1}, \mathbf{X}_{t-L+2}, ..., \mathbf{X}_t)$, where H is the horizon length of the future time series. Therefore, we can formulate the multivariate time series forecasting problem as follows:

$$(\hat{\mathbf{X}}_{t+1}, \hat{\mathbf{X}}_{t+2}, ..., \hat{\mathbf{X}}_{t+H}) = \mathcal{F}_\Theta(\mathbf{X}_{t-L+1}, \mathbf{X}_{t-L+2}, ..., \mathbf{X}_t) \qquad (14)$$

where $\Theta$ is the parameters of the forecasting model $\mathcal{F}$.

## A.2  TREND AND SEASONALITY VARIATIONS

In order to better describe the properties of the datasets, we need to calculate the trend variation and seasonality variation.

**Trend Variation.** To capture the global trend change, we calculate the average value of different regions of the dataset. With a time series dataset $\mathbf{X} \in \mathbb{R}^{N \times L}$, where $N$ is the number of inputs and $L$ is the lookback length, we first split it into $\mathbf{X}_{train}$, $\mathbf{X}_{val}$, and $\mathbf{X}_{test}$ in chronological order, representing the training dataset, validation dataset, and test dataset, respectively. Then, the trend variation is calculated as follows:

$$\text{Trend Variation} = \left| \frac{\text{Mean}_N(\mathbf{X}_{train}) - \text{Mean}_N(\mathbf{X}_{val,test})}{\text{Mean}_N(\mathbf{X}_{train})} \right| \qquad (15)$$

where $\mathbf{X}_{val,test}$ represents the concatenation of the validation set and the test set. It should be noted that in order to obtain relative results between different datasets, the trend changes need to be normalized by dividing by the mean of the training dataset.

**Seasonality Variation.** We evaluate seasonal changes by analyzing the Fourier frequency changes of all input instances. Given an input $\mathbf{X} \in \mathbb{R}^{N_i \times L}$, where $N_i$ is the number of inputs and $L$ is the lookback length. We first obtain the FFT results of all inputs, denoted as $Z \in \mathbb{C}^{N_i \times L}$. Then, we calculate the variance between different inputs and normalize the variance by dividing by the mean of each input, as follows:

$$\text{Seasonality Variation} = \frac{\text{Var}_{N_i}[\text{Amp}(Z)]}{\text{Mean}_L(X)} \qquad (16)$$

where the subscripts indicate the dimension of the operation process.

## A.3  FAST FOURIER TRANSFORM (FFT)

In time series forecasting, Fast Fourier Transform (FFT) is often used for frequency domain analysis Brigham (1988). Here, we perform FFT decomposition on the time series $\mathbf{X}_t (t = 0, 1, \ldots, L-1)$ of length $L$ and obtain the frequency domain coefficients:

$$\mathbf{X}_k = \sum_{t=0}^{L-1} x_t \cdot e^{-i2\pi kt/L}, \qquad (17)$$

where $k = 0, 1, \ldots, L-1$ and $\mathbf{X}_k$ is a complex number consisting of amplitude $\mathbf{A}_k$ and phase $\phi_k$:

$$\mathbf{X}_k = \mathbf{A}_k e^{i\phi_k}, \qquad (18)$$
$$\mathbf{A}_k = |\mathbf{X}_k|, \qquad (19)$$
$$\phi_k = \arg(\mathbf{X}_k). \qquad (20)$$

where $|\cdot|$ represents the absolute value operation and $\arg(\cdot)$ is the argument function of a complex number, which is used to calculate the phase angle of a complex number in the complex plane.

## A.4 DISCRETE FOURIER TRANSFORM (DFT) AND IDFT PROCESS

Based on Section A.3, we introduce the Discrete Fourier Transform (DFT) process and Inverse Discrete Fourier Transform (IDFT) process that can be implemented by Fast Fourier Transform (FFT) Brigham (1988). Given a multivariate time series input $\mathbf{X}$, we perform a 1-dim Fourier transform on each dimension $\mathrm{X}^{(i)}$ separately, so we illustrate it in vector form. For a discrete time series vector $\mathbf{X} \in \mathbb{R}^L$ with the lookback length of $L$, we transform it to the Fourier domain by applying a 1-dim DFT, and then we can also transform it back to the Fourier domain using a 1-dim IDFT, which is defined as:

$$\mathrm{DFT} : \mathbf{Z}[\omega] = \sum_{t=0}^{L-1} \mathbf{X}[t] \cdot e^{-2\pi i \frac{\omega t}{L}} \tag{21}$$

$$\mathrm{IDFT} : \mathbf{X}[t] = \frac{1}{L} \sum_{\omega=0}^{T-1} \mathbf{Z}[\omega] \cdot e^{2\pi i \frac{\omega t}{L}} \tag{22}$$

where $\omega$ is the current frequency, $t$ is the current time step, and $\mathbf{Z}$ is the result of the Fourier transform, which is a complex vector containing real and imaginary parts. Its amplitude and phase can be calculated as follows:

$$\mathrm{Mag} : \mathbf{a}[\omega] = \frac{\sqrt{\mathrm{Re}(\mathbf{Z}[\omega])^2 + \mathrm{Im}(\mathbf{Z}[\omega])^2}}{L} \tag{23}$$

$$\mathrm{Pha} : \mathbf{p}[\omega] = \mathrm{atan}\,2(\mathrm{Im}(\mathbf{Z}[\omega]), \mathrm{Re}(\mathbf{Z}[\omega])) \tag{24}$$

where $\mathrm{Im}(\mathbf{Z}[\cdot])$ and $\mathrm{Re}(\mathbf{Z}[\cdot])$ represent the imaginary and real parts of the complex number, respectively, and $\mathrm{atan}\,2$ is the two-argument form of $arctan$.

## A.5 FREQUENCY ADAPTIVE NORMALIZATION (FAN) METHOD

In this section, we briefly introduce the frequency adaptive normalization (FAN) method Ye et al. (2024). Please refer to the original paper Ye et al. (2024) for specific related functions and variable names.

At each time step, FAN Ye et al. (2024) first removes the first $K$ dominant components in the frequency domain for each input instance. This process is called frequency residual learning (FRL), and then removes $\mathbf{X}_{non}$ from the original sequence to obtain the stationary component $\mathbf{X}_{res}$:

$$\mathbf{Z} = \mathrm{DFT}(\mathbf{X}), \tag{25}$$

$$\mathcal{K} = \mathrm{TopK}(\mathrm{Amp}(\mathbf{Z})), \tag{26}$$

$$\mathbf{X}_{non} = \mathrm{IDFT}(\mathrm{Filter}(\mathcal{K}, \mathbf{Z})), \tag{27}$$

$$\mathbf{X}_{res} = \mathbf{X} - \mathbf{X}_{non}, \tag{28}$$

The DFT and IDFT processes can be implemented using Fast Fourier Transform (FFT). Afterwards, the prediction backbone $g_\theta$ uses the stationary component $\mathbf{X}_{res}$ to forecast the stationary part of the output $\widehat{Y}_{res}$ and then reintegrates the removed non-stationary information into the output:

$$\hat{\mathbf{Y}}_{res} = g_\theta(\mathbf{X}_{res}), \tag{29}$$

$$\hat{\mathbf{Y}} = \hat{\mathbf{Y}}_{res} + \hat{\mathbf{Y}}_{non}, \tag{30}$$

Here, a simple multi-layer perceptron (MLP) model $q_\phi$ is used to directly predict the future values of the composite top $K$ frequency components for $D$ dimensions:

$$\hat{\mathbf{Y}}_{non} = q_\phi(\mathbf{X}_{non}, \mathbf{X}) \tag{31}$$

$$= \mathbf{W}_3 \, \mathrm{ReLU}(\mathbf{W}_2 \, \mathrm{Concat}(\mathrm{ReLU}(\mathbf{W}_1 \mathbf{X}_{non}), \mathbf{X})) \tag{32}$$

The above is a brief introduction to the preparation work for this paper. For more details about the FAN method Ye et al. (2024), please refer to the original paper Ye et al. (2024).

# B IMPLEMENTATION DETAILS

In this section, we will introduce the specific details of the datasets and the evaluation metrics to help readers better reproduce the experimental results of this paper.

## B.1 DATASETS DETAILS

We use 8 widely-used real-world datasets in the time series field, namely the ETT (Electric Transformer Temperature) dataset Zhou et al. (2021), which records the oil temperature and load of power transformers for 2 years from July 2016 to July 2018. The dataset contains 4 subsets, of which (1) ETTh1 and (2) ETTh2 are sampled every hour, and (3) ETTm1 and (4) ETTm2 are sampled every 15 minutes; (5) Electricity, which contains the electricity consumption of 321 customers every 15 minutes for 3 years from July 2016 to July 2019; (6) Exchange Rate, which records the daily exchange rates of 8 countries for 26 years from 1990 to 2016. (7) Traffic, which contains hourly traffic flow on San Francisco highways recorded by 862 sensors for 1 year from 2015 to 2016; (8) Weather, which consists of 21 meteorological indicators, including air temperature and humidity data collected every 10 minutes in 2021. For more detailed properties and characteristics of the datasets, please refer to Table 6.

Table 6: The detailed descriptions of the datasets.

| Datasets | $Dim$ | Dataset Size | Frequency | $K$ | $TV$ | $SV$ | Information |
|----------|-------|--------------|-----------|-----|------|------|-------------|
| ETTh1 | 7 | (8545, 2881, 2881) | 1 Hour | 4 | 3.839 | 3.690 | Temperature |
| ETTh2 | 7 | (8545, 2881, 2881) | 1 Hour | 3 | 0.154 | 1.013 | Temperature |
| ETTm1 | 7 | (34465, 11521, 11521) | 15 Minutes | 11 | 0.030 | 3.330 | Temperature |
| ETTm2 | 7 | (34465, 11521, 11521) | 15 Minutes | 5 | 0.196 | 1.648 | Temperature |
| Electricity | 321 | (18317, 2633, 5261) | 1 Hour | 2 | 0.249 | 0.435 | Electricity |
| Exchange | 8 | (5120, 665, 1422) | 1 Day | 3 | 0.242 | 2.645 | Exchange Rate |
| Traffic | 862 | (12185, 1757, 3509) | 1 Hour | 30 | 0.068 | 14.225 | Transportation |
| Weather | 21 | (36792, 5271, 10540) | 10 Minutes | 2 | 0.028 | 0.387 | Weather |

As shown in Table 6, $Dim$ represents the dimension of the dataset, which is the number of variables, and the dataset size is listed as (Train, Validation, Test). $K$ is the hyperparameter of the top $K$ amplitude signals proposed in the FAN method Ye et al. (2024). For more details on the hyperparameter $K$, please refer to the original paper of FAN Ye et al. (2024). Furthermore, $TV$ and $SV$ represent trend variation and seasonality variation, respectively, mentioned in Appendix A.2.

## B.2 METRICS DETAILS

Regarding metrics, we use the mean square error (MSE) and mean absolute error (MAE) as evaluation metrics for time series forecasting, which are calculated as follows:

$$\text{MSE} = \frac{1}{H} \sum_{i=1}^{H} (X_i - \widehat{X}_i)^2 \tag{33}$$

$$\text{MAE} = \frac{1}{H} \sum_{i=1}^{H} |X_i - \widehat{X}_i| \tag{34}$$

where $X_i, \widehat{X}_i \in \mathbb{R}$ are the ground truth and prediction results of the $i^{th}$ time point in the future and $N$ is the total number of future time points.

## C ALGORITHMIC DETAILS OF MODEL DESIGN

In this section, in order to help readers understand the core idea of this paper more clearly, we introduce the specific algorithmic processes of the two major innovations proposed in this paper, sliding window adaptive normalization (SWAN) and statistical prediction Module (SPM).

### C.1 SLIDING WINDOW ADAPTIVE NORMALIZATION (SWAN)

Regarding the sliding window adaptive normalization (SWAN) module, we describe the specific algorithm flow of dynamic optimal window size selection and sliding window adaptive normalization in Algorithm 1 and Algorithm 2.

---

**Algorithm 1:** Dynamic Optimal Window Size Selection

---

**Input:** The set of the candidate window size $W \in \phi_w$; the time series data $X$; and the lookback length $L$

**Output:** The optimal window size $W_{optimal}$

1 **Initialisation:** Initialize the candidate window size set $\phi_w = \{12, 24, 48\}$

2 **while** $W \in \phi_w$ **do**

3     Padding the sequence with a size of $\frac{W}{2}$

4     **for** $i \leftarrow 1$ **to** $L$ **do**

5        $\sigma_{window}(i) = \sqrt{\frac{1}{W} \sum_{j=i}^{W}(X_j - \mu_{window}(i))^2}$

6     **end for**

7     $\mu_\sigma \leftarrow \frac{1}{L} \sum_{i=1}^{L} \sigma_{window}(i)$

8     $\sigma_{window} \leftarrow \sqrt{\frac{1}{L} \sum_{i=1}^{L}(\sigma_{window}(i) - \mu_\sigma)^2}$

9 **end while**

10 **return** $W_{optimal} \leftarrow arg\ min_{\phi_w}\ \sigma_{window}$

---

---

**Algorithm 2:** Sliding Window Adaptive Normalization

---

**Input:** The optimal window size $W_{optimal}$; the original time series data $X$; and the lookback length $L$

**Output:** The normalized time series data $X_{stat}$

1 **Initialisation:** Define the set of means $\phi_\mu$ and the set of standard deviations $\phi_\sigma$ containing the statistics of each window, and the set of the normalized sequence $\phi_{X_{norm}}$

2 Padding the sequence with a size of $\frac{W_{optimal}}{2}$

3 **for** $i = 1$ **to** $L$ **do**

4     $\mu_{window}(i) \leftarrow \frac{1}{W} \sum_{j=i}^{W} X_j$

5     $\sigma_{window}(i) \leftarrow \sqrt{\frac{1}{W} \sum_{j=i}^{W}(X_j - \mu_{window}(i))^2}$

6     $X_{stat}(i) \leftarrow \frac{X(i) - \mu_{window}(i)}{\sigma_{window}(i) + \varepsilon}$

7     $\phi_\mu \leftarrow \phi_\mu \cup \{\mu_{window}(i)\}$

8     $\phi_\sigma \leftarrow \phi_\sigma \cup \{\sigma_{window}(i)\}$

9     $\phi_{X_{stat(i)}} \leftarrow \phi_{X_{stat(i)}} \cup \{X_{stat}(i)\}$

10 **end for**

11 **return** $X_{stat} \leftarrow \phi_{X_{stat(i)}}$

---

## C.2 STATISTICAL PREDICTION MODULE (SPM)

For the statistical prediction module (SPM), we describe its detailed process in Algorithm 3.

---

**Algorithm 3:** Statistical Prediction Module

---

**Input:** The statistics $\mu$ and $\sigma$ from the sets $\phi_\mu$ and $\phi_\sigma$ calculated in Algorithm 2; the original time series data $X$; the stationary part results $\hat{Y}^{stat}$ predicted by the backbone network with the input $X_{stat}$

**Output:** The predicted stationary component $\widehat{Y}_{res}$

1 **Initialisation:** Initialize the network structure of $f_1$ and $f_2$, which contain 1 and 2 linear layers respectively, and the ReLU activation function, where $L$ and $H$ represent the lookback and horizon lengths respectively

2 $h_\mu \leftarrow \mathrm{ReLU}(\mathrm{Linear}_{f_1,\ L \times 256}(\mu))$

3 $inp_\mu \leftarrow \mathrm{Concat}(h_\mu, X)$

4 $h_\sigma \leftarrow \mathrm{ReLU}(\mathrm{Linear}_{f_1,\ L \times 256}(\sigma))$

5 $inp_\sigma \leftarrow \mathrm{Concat}(h_\sigma, X)$

6 $h_\mu \leftarrow \mathrm{ReLU}(\mathrm{Linear}_{f_2,\ (256+L) \times 512}(inp_\mu))$

7 $\widehat{\mu} \leftarrow \mathrm{Linear}_{f_2,\ 512 \times H}(h_\mu)$

8 $h_\sigma \leftarrow \mathrm{ReLU}(\mathrm{Linear}_{f_2,\ (256+L) \times 512}(inp_\sigma))$

9 $\widehat{\sigma} \leftarrow \mathrm{Linear}_{f_2,\ 512 \times H}(h_\sigma)$

10 **return** $\hat{Y}^{res} \leftarrow \hat{Y}^{stat} \cdot \widehat{\sigma} + \widehat{\mu}$

---

# D  ADDITIONAL RESULTS

## D.1  ADDITIONAL EXPERIMENTS RESULTS ON OTHER BACKBONES

We present the experimental results of incorporating our Dual-AN method on 5 datasets with 3 state-of-the-art backbones: (1) MLP-based WPMixer Murad et al. (2025); (2) Transformer-based iTransformer Liu et al. (2023a); (3) CNN-based MICN Wang et al. (2023), in Tables 7, 8, and 9.

Table 7: Full results of the WPMixer backbone with and without Dual-AN. The best results are highlighted in **bold**.

| Models | Metrics | WPMixer MAE | WPMixer MSE | +Dual-AN MAE | +Dual-AN MSE |
|--------|---------|-------------|-------------|--------------|--------------|
| ETTh1 | 96 | 0.430 ± 0.002 | 0.374 ± 0.003 | **0.426 ± 0.002** | **0.363 ± 0.002** |
| | 168 | 0.460 ± 0.001 | 0.411 ± 0.002 | **0.449 ± 0.004** | **0.390 ± 0.006** |
| | 336 | **0.485 ± 0.002** | 0.456 ± 0.003 | 0.487 ± 0.003 | **0.446 ± 0.004** |
| | 720 | 0.574 ± 0.007 | 0.609 ± 0.012 | **0.571 ± 0.002** | **0.571 ± 0.003** |
| ETTh2 | 96 | 0.239 ± 0.002 | 0.115 ± 0.002 | **0.239 ± 0.001** | **0.113 ± 0.000** |
| | 168 | 0.258 ± 0.002 | 0.134 ± 0.002 | **0.256 ± 0.005** | **0.130 ± 0.002** |
| | 336 | 0.275 ± 0.006 | 0.151 ± 0.005 | **0.271 ± 0.005** | **0.143 ± 0.004** |
| | 720 | 0.302 ± 0.007 | 0.188 ± 0.008 | **0.284 ± 0.002** | **0.160 ± 0.001** |
| ETTm2 | 96 | 0.200 ± 0.000 | 0.079 ± 0.000 | **0.198 ± 0.001** | **0.077 ± 0.001** |
| | 168 | 0.220 ± 0.001 | 0.094 ± 0.000 | **0.218 ± 0.001** | **0.092 ± 0.000** |
| | 336 | 0.245 ± 0.001 | 0.118 ± 0.001 | **0.242 ± 0.001** | **0.115 ± 0.001** |
| | 720 | 0.270 ± 0.002 | 0.150 ± 0.001 | **0.264 ± 0.000** | **0.139 ± 0.001** |
| Exchange | 96 | **0.165 ± 0.001** | **0.054 ± 0.001** | 0.169 ± 0.001 | **0.054 ± 0.001** |
| | 168 | **0.214 ± 0.001** | **0.087 ± 0.001** | 0.222 ± 0.004 | 0.091 ± 0.002 |
| | 336 | 0.311 ± 0.004 | 0.177 ± 0.005 | **0.283 ± 0.005** | **0.151 ± 0.003** |
| | 720 | 0.483 ± 0.006 | 0.384 ± 0.007 | **0.432 ± 0.009** | **0.318 ± 0.013** |
| Traffic | 96 | 0.354 ± 0.003 | 0.440 ± 0.003 | **0.324 ± 0.001** | **0.391 ± 0.001** |
| | 168 | 0.353 ± 0.002 | 0.446 ± 0.003 | **0.328 ± 0.001** | **0.405 ± 0.001** |
| | 336 | 0.363 ± 0.002 | 0.467 ± 0.001 | **0.340 ± 0.001** | **0.429 ± 0.002** |
| | 720 | 0.387 ± 0.004 | 0.497 ± 0.003 | **0.365 ± 0.000** | **0.463 ± 0.000** |

As shown in Table 7, 8, and 9, after adding the Dual-AN method to the WPMixer, iTransformer, and MICN backbones, the average MAE/MSE ratios across all the 5 datasets decrease by 3.40%/7.37%, 9.78%/18.31%, and 4.81%/6.87%, respectively.

Table 8: Full results of the iTransformer backbone with and without Dual-AN. The best results are highlighted in **bold**.

| Models | | iTransformer | | +Dual-AN | |
|---|---|---|---|---|---|
| Metrics | | MAE | MSE | MAE | MSE |
| ETTh1 | 96 | 0.444 ± 0.005 | 0.378 ± 0.007 | **0.426 ± 0.001** | **0.362 ± 0.000** |
| | 168 | 0.472 ± 0.009 | 0.413 ± 0.012 | **0.449 ± 0.002** | **0.390 ± 0.003** |
| | 336 | 0.533 ± 0.015 | 0.511 ± 0.023 | **0.486 ± 0.003** | **0.441 ± 0.005** |
| | 720 | 0.640 ± 0.021 | 0.669 ± 0.043 | **0.569 ± 0.002** | **0.574 ± 0.006** |
| ETTh2 | 96 | 0.255 ± 0.004 | 0.122 ± 0.002 | **0.237 ± 0.001** | **0.111 ± 0.001** |
| | 168 | 0.271 ± 0.009 | 0.141 ± 0.006 | **0.252 ± 0.002** | **0.128 ± 0.001** |
| | 336 | 0.300 ± 0.020 | 0.167 ± 0.017 | **0.264 ± 0.001** | **0.139 ± 0.001** |
| | 720 | 0.362 ± 0.041 | 0.482 ± 0.041 | **0.279 ± 0.002** | **0.155 ± 0.001** |
| ETTm2 | 96 | 0.203 ± 0.005 | 0.078 ± 0.003 | **0.199 ± 0.000** | **0.078 ± 0.000** |
| | 168 | 0.226 ± 0.005 | 0.094 ± 0.003 | **0.219 ± 0.000** | **0.093 ± 0.000** |
| | 336 | 0.247 ± 0.005 | 0.114 ± 0.003 | **0.243 ± 0.001** | **0.114 ± 0.000** |
| | 720 | 0.277 ± 0.004 | 0.144 ± 0.004 | **0.264 ± 0.000** | **0.139 ± 0.000** |
| Exchange | 96 | 0.227 ± 0.021 | 0.093 ± 0.015 | **0.168 ± 0.001** | **0.054 ± 0.001** |
| | 168 | 0.270 ± 0.023 | 0.131 ± 0.020 | **0.218 ± 0.002** | **0.090 ± 0.001** |
| | 336 | 0.390 ± 0.050 | 0.262 ± 0.063 | **0.294 ± 0.001** | **0.161 ± 0.001** |
| | 720 | 0.512 ± 0.096 | 0.480 ± 0.166 | **0.409 ± 0.016** | **0.291 ± 0.017** |
| Traffic | 96 | 0.320 ± 0.013 | **0.371 ± 0.017** | **0.319 ± 0.000** | 0.388 ± 0.001 |
| | 168 | 0.337 ± 0.001 | **0.408 ± 0.001** | **0.330 ± 0.000** | 0.408 ± 0.000 |
| | 336 | 0.350 ± 0.001 | 0.432 ± 0.001 | **0.335 ± 0.000** | **0.427 ± 0.000** |
| | 720 | 0.376 ± 0.002 | 0.469 ± 0.002 | **0.357 ± 0.000** | **0.458 ± 0.000** |

Table 9: Full results of the MICN backbone with and without Dual-AN. The best results are highlighted in **bold**.

| Models | | MICN | | +Dual-AN | |
|---|---|---|---|---|---|
| Metrics | | MAE | MSE | MAE | MSE |
| ETTh1 | 96 | 0.454 ± 0.001 | 0.387 ± 0.002 | **0.420 ± 0.002** | **0.355 ± 0.002** |
| | 168 | 0.485 ± 0.003 | 0.433 ± 0.004 | **0.449 ± 0.003** | **0.388 ± 0.004** |
| | 336 | 0.551 ± 0.004 | 0.533 ± 0.007 | **0.495 ± 0.003** | **0.453 ± 0.004** |
| | 720 | 0.609 ± 0.003 | 0.626 ± 0.005 | **0.580 ± 0.003** | **0.576 ± 0.005** |
| ETTh2 | 96 | 0.239 ± 0.003 | **0.110 ± 0.002** | **0.237 ± 0.001** | 0.111 ± 0.001 |
| | 168 | 0.259 ± 0.002 | 0.128 ± 0.002 | **0.248 ± 0.003** | **0.124 ± 0.001** |
| | 336 | 0.287 ± 0.002 | 0.148 ± 0.002 | **0.261 ± 0.003** | **0.135 ± 0.002** |
| | 720 | 0.338 ± 0.004 | 0.200 ± 0.005 | **0.283 ± 0.002** | **0.155 ± 0.001** |
| ETTm2 | 96 | 0.195 ± 0.001 | 0.074 ± 0.000 | **0.192 ± 0.001** | **0.073 ± 0.001** |
| | 168 | 0.215 ± 0.001 | 0.088 ± 0.000 | **0.212 ± 0.000** | **0.088 ± 0.000** |
| | 336 | **0.235 ± 0.001** | **0.106 ± 0.001** | 0.239 ± 0.003 | 0.111 ± 0.003 |
| | 720 | 0.267 ± 0.002 | **0.136 ± 0.002** | **0.264 ± 0.001** | 0.138 ± 0.000 |
| Exchange | 96 | 0.171 ± 0.003 | 0.056 ± 0.002 | **0.169 ± 0.002** | **0.055 ± 0.001** |
| | 168 | **0.217 ± 0.002** | **0.088 ± 0.002** | 0.224 ± 0.006 | 0.092 ± 0.004 |
| | 336 | 0.309 ± 0.002 | 0.172 ± 0.002 | **0.298 ± 0.007** | **0.162 ± 0.004** |
| | 720 | 0.495 ± 0.022 | 0.417 ± 0.034 | **0.428 ± 0.023** | **0.319 ± 0.028** |
| Traffic | 96 | 0.323 ± 0.003 | 0.380 ± 0.006 | **0.320 ± 0.002** | **0.379 ± 0.002** |
| | 168 | 0.334 ± 0.002 | **0.402 ± 0.003** | **0.325 ± 0.004** | 0.402 ± 0.006 |
| | 336 | 0.345 ± 0.006 | 0.427 ± 0.011 | **0.342 ± 0.001** | **0.430 ± 0.001** |
| | 720 | 0.358 ± 0.007 | 0.446 ± 0.006 | **0.351 ± 0.001** | **0.433 ± 0.001** |

### D.2 COMPLETE EXPERIMENTAL RESULTS

Due to the space limitation of the main text, we place the complete experimental results of the 3 backbone models with and without Dual-AN on all 8 datasets in Table 10.

Table 10: Full results of the main experiments with and without Dual-AN. The best results are highlighted in **bold**.

| Models | | DLinear | | +Dual-AN | | Informer | | +Dual-AN | | SCINet | | +Dual-AN | |
|---|---|---|---|---|---|---|---|---|---|---|---|---|---|
| Metrics | | MAE | MSE | MAE | MSE | MAE | MSE | MAE | MSE | MAE | MSE | MAE | MSE |
| ETTh1 | 96 | **0.424** | 0.368 | 0.425 | **0.363** | 0.598 | 0.646 | **0.421** | **0.357** | 0.461 | 0.409 | **0.420** | **0.356** |
| | 168 | **0.449** | 0.398 | 0.453 | **0.396** | 0.694 | 0.863 | **0.446** | **0.386** | 0.518 | 0.489 | 0.452 | **0.396** |
| | 336 | **0.485** | 0.448 | 0.487 | **0.446** | 0.738 | 0.950 | 0.493 | 0.452 | 0.574 | 0.582 | 0.493 | 0.450 |
| | 720 | **0.561** | **0.558** | 0.573 | 0.568 | 0.823 | 1.106 | 0.579 | 0.589 | 0.645 | 0.707 | 0.581 | 0.581 |
| ETTh2 | 96 | 0.237 | **0.110** | **0.236** | **0.110** | 0.298 | 0.160 | 0.238 | 0.111 | 0.264 | 0.128 | **0.237** | 0.112 |
| | 168 | 0.254 | 0.127 | **0.250** | **0.125** | 0.331 | 0.191 | 0.252 | 0.127 | 0.292 | 0.156 | **0.249** | **0.125** |
| | 336 | 0.271 | **0.138** | **0.264** | **0.138** | 0.347 | 0.208 | 0.276 | 0.147 | 0.305 | 0.167 | **0.262** | 0.137 |
| | 720 | 0.316 | 0.179 | **0.280** | **0.157** | 0.413 | 0.291 | 0.337 | 0.208 | 0.339 | 0.201 | 0.284 | 0.156 |
| ETTm1 | 96 | **0.380** | **0.310** | 0.394 | 0.334 | 0.514 | 0.520 | 0.401 | 0.353 | 0.421 | 0.355 | 0.395 | 0.343 |
| | 168 | **0.408** | **0.354** | 0.414 | 0.360 | 0.563 | 0.600 | 0.422 | 0.377 | 0.446 | 0.399 | 0.414 | 0.360 |
| | 336 | **0.446** | **0.416** | 0.455 | 0.421 | 0.612 | 0.690 | 0.459 | 0.429 | 0.489 | 0.464 | 0.454 | 0.421 |
| | 720 | **0.488** | **0.471** | 0.492 | 0.474 | 0.697 | 0.849 | 0.494 | 0.477 | 0.553 | 0.563 | 0.487 | 0.465 |
| ETTm2 | 96 | 0.203 | 0.080 | **0.199** | **0.078** | 0.226 | 0.091 | **0.199** | 0.079 | 0.206 | 0.079 | **0.199** | **0.078** |
| | 168 | 0.220 | **0.093** | **0.219** | **0.093** | 0.251 | 0.112 | 0.220 | **0.093** | 0.226 | 0.094 | **0.219** | **0.093** |
| | 336 | 0.245 | 0.114 | **0.242** | **0.113** | 0.283 | 0.140 | 0.245 | 0.114 | 0.262 | 0.122 | **0.242** | **0.113** |
| | 720 | 0.270 | 0.142 | **0.264** | **0.139** | 0.347 | 0.212 | 0.277 | 0.147 | 0.297 | 0.153 | **0.264** | **0.139** |
| Electricity | 96 | 0.277 | 0.195 | **0.265** | **0.181** | 0.376 | 0.277 | **0.244** | **0.148** | 0.296 | 0.188 | **0.254** | **0.159** |
| | 168 | 0.272 | 0.183 | **0.265** | **0.176** | 0.371 | 0.269 | **0.254** | **0.159** | 0.306 | 0.196 | **0.256** | **0.160** |
| | 336 | 0.294 | 0.197 | **0.285** | **0.190** | 0.377 | 0.273 | **0.270** | **0.166** | 0.330 | 0.214 | **0.272** | **0.169** |
| | 720 | 0.333 | 0.233 | **0.320** | **0.223** | 0.401 | 0.311 | **0.302** | **0.191** | 0.352 | 0.240 | **0.303** | **0.194** |
| Exchange | 96 | **0.164** | **0.052** | 0.167 | 0.053 | 0.532 | 0.412 | **0.168** | **0.055** | 0.218 | 0.085 | **0.167** | **0.053** |
| | 168 | 0.219 | 0.090 | **0.215** | **0.087** | 0.582 | 0.491 | **0.217** | **0.089** | 0.266 | 0.126 | **0.215** | **0.087** |
| | 336 | **0.288** | **0.155** | 0.291 | 0.158 | 0.721 | 0.847 | **0.295** | **0.164** | 0.337 | 0.203 | **0.290** | **0.156** |
| | 720 | 0.453 | 0.352 | **0.398** | **0.283** | 0.889 | 1.210 | **0.431** | **0.350** | 0.502 | 0.430 | **0.427** | **0.312** |
| Traffic | 96 | 0.387 | 0.504 | **0.334** | **0.403** | 0.350 | 0.428 | **0.323** | **0.386** | 0.399 | 0.471 | **0.325** | **0.393** |
| | 168 | 0.588 | 0.804 | **0.333** | **0.413** | 0.366 | 0.457 | **0.320** | **0.393** | 0.377 | 0.443 | **0.328** | **0.408** |
| | 336 | 0.380 | 0.504 | **0.345** | **0.436** | 0.414 | 0.555 | **0.336** | **0.425** | 0.384 | 0.459 | **0.345** | **0.436** |
| | 720 | 0.407 | 0.532 | **0.368** | **0.469** | 0.656 | 1.002 | **0.356** | **0.448** | 0.401 | 0.490 | **0.368** | **0.469** |
| Weather | 96 | 0.249 | **0.180** | **0.220** | 0.181 | 0.299 | 0.221 | **0.210** | **0.172** | 0.265 | 0.199 | **0.211** | **0.170** |
| | 168 | 0.284 | 0.237 | **0.259** | **0.218** | 0.363 | 0.320 | **0.250** | **0.211** | 0.305 | 0.245 | **0.252** | **0.209** |
| | 336 | 0.344 | 0.304 | **0.298** | **0.278** | 0.439 | 0.437 | **0.301** | **0.270** | 0.341 | 0.310 | **0.293** | **0.271** |
| | 720 | 0.380 | 0.358 | **0.346** | **0.343** | 0.496 | 0.524 | **0.366** | **0.349** | 0.383 | 0.371 | **0.331** | **0.329** |

In addition, we report the full results of comparative experiments with existing normalization methods on all 8 datasets in Table 11. For the model parameter experiments on lookback and horizon lengths, we place the complete experimental results in Table 12.

Table 11: Full results of MAE and MSE performance compared with other normalization methods. The best performance is highlighted in red and the second best performance is underlined.

| | | | DLinear | | | | | Informer | | | | | SCINet | | | | |
|---|---|---|---|---|---|---|---|---|---|---|---|---|---|---|---|---|---|
| **Models** | | | Dual-AN | FAN | SAN | Dish-TS | RevIN | Dual-AN | FAN | SAN | Dish-TS | RevIN | Dual-AN | FAN | SAN | Dish-TS | RevIN |
| ETTh1 | 96 | MAE | 0.425 | 0.426 | 0.432 | 0.433 | 0.428 | 0.421 | 0.434 | 0.498 | 0.556 | 0.521 | 0.420 | 0.427 | 0.431 | 0.438 | 0.438 |
| | | MSE | 0.363 | 0.362 | 0.370 | 0.375 | 0.375 | 0.357 | 0.367 | 0.466 | 0.549 | 0.517 | 0.356 | 0.362 | 0.370 | 0.382 | 0.380 |
| | 168 | MAE | 0.453 | 0.452 | 0.460 | 0.454 | 0.464 | 0.446 | 0.465 | 0.514 | 0.601 | 0.539 | 0.452 | 0.454 | 0.459 | 0.476 | 0.470 |
| | | MSE | 0.396 | 0.393 | 0.404 | 0.405 | 0.416 | 0.386 | 0.407 | 0.485 | 0.642 | 0.531 | 0.396 | 0.395 | 0.404 | 0.430 | 0.425 |
| | 336 | MAE | 0.487 | 0.484 | 0.504 | 0.505 | 0.501 | 0.493 | 0.507 | 0.627 | 0.662 | 0.642 | 0.493 | 0.487 | 0.502 | 0.539 | 0.490 |
| | | MSE | 0.446 | 0.435 | 0.463 | 0.475 | 0.476 | 0.452 | 0.467 | 0.702 | 0.753 | 0.735 | 0.450 | 0.439 | 0.461 | 0.522 | 0.462 |
| | 720 | MAE | 0.573 | 0.572 | 0.584 | 0.590 | 0.598 | 0.579 | 0.602 | 0.689 | 0.739 | 0.763 | 0.581 | 0.572 | 0.579 | 0.604 | 0.584 |
| | | MSE | 0.568 | 0.574 | 0.579 | 0.603 | 0.641 | 0.589 | 0.617 | 0.845 | 0.914 | 0.968 | 0.581 | 0.575 | 0.580 | 0.622 | 0.620 |
| ETTh2 | 96 | MAE | 0.236 | 0.234 | 0.237 | 0.237 | 0.239 | 0.238 | 0.256 | 0.272 | 0.330 | 0.309 | 0.237 | 0.239 | 0.238 | 0.265 | 0.241 |
| | | MSE | 0.110 | 0.108 | 0.112 | 0.111 | 0.117 | 0.111 | 0.124 | 0.138 | 0.196 | 0.178 | 0.112 | 0.112 | 0.113 | 0.132 | 0.115 |
| | 168 | MAE | 0.250 | 0.251 | 0.252 | 0.255 | 0.255 | 0.252 | 0.269 | 0.296 | 0.361 | 0.317 | 0.249 | 0.255 | 0.252 | 0.281 | 0.263 |
| | | MSE | 0.125 | 0.126 | 0.128 | 0.129 | 0.135 | 0.127 | 0.138 | 0.159 | 0.234 | 0.189 | 0.125 | 0.130 | 0.127 | 0.152 | 0.140 |
| | 336 | MAE | 0.264 | 0.263 | 0.264 | 0.269 | 0.273 | 0.276 | 0.300 | 0.310 | 0.375 | 0.334 | 0.262 | 0.269 | 0.263 | 0.297 | 0.275 |
| | | MSE | 0.138 | 0.132 | 0.137 | 0.138 | 0.152 | 0.147 | 0.162 | 0.176 | 0.254 | 0.205 | 0.137 | 0.142 | 0.136 | 0.165 | 0.151 |
| | 720 | MAE | 0.280 | 0.281 | 0.286 | 0.288 | 0.303 | 0.337 | 0.378 | 0.416 | 0.436 | 0.354 | 0.284 | 0.284 | 0.304 | 0.321 | 0.305 |
| | | MSE | 0.157 | 0.158 | 0.159 | 0.165 | 0.193 | 0.208 | 0.256 | 0.332 | 0.350 | 0.223 | 0.156 | 0.159 | 0.179 | 0.190 | 0.190 |
| ETTm1 | 96 | MAE | 0.394 | 0.394 | 0.386 | 0.407 | 0.383 | 0.401 | 0.389 | 0.401 | 0.457 | 0.446 | 0.395 | 0.394 | 0.389 | 0.415 | 0.436 |
| | | MSE | 0.334 | 0.334 | 0.311 | 0.356 | 0.317 | 0.353 | 0.322 | 0.330 | 0.445 | 0.420 | 0.343 | 0.333 | 0.321 | 0.357 | 0.423 |
| | 168 | MAE | 0.414 | 0.416 | 0.416 | 0.421 | 0.435 | 0.422 | 0.417 | 0.443 | 0.496 | 0.470 | 0.414 | 0.415 | 0.422 | 0.442 | 0.454 |
| | | MSE | 0.360 | 0.364 | 0.354 | 0.373 | 0.390 | 0.377 | 0.362 | 0.393 | 0.496 | 0.457 | 0.360 | 0.363 | 0.367 | 0.414 | 0.430 |
| | 336 | MAE | 0.455 | 0.456 | 0.458 | 0.459 | 0.480 | 0.459 | 0.462 | 0.492 | 0.536 | 0.524 | 0.454 | 0.456 | 0.454 | 0.481 | 0.490 |
| | | MSE | 0.421 | 0.423 | 0.415 | 0.433 | 0.463 | 0.429 | 0.425 | 0.492 | 0.552 | 0.525 | 0.421 | 0.423 | 0.415 | 0.467 | 0.486 |
| | 720 | MAE | 0.492 | 0.493 | 0.497 | 0.501 | 0.530 | 0.494 | 0.506 | 0.545 | 0.608 | 0.597 | 0.487 | 0.495 | 0.498 | 0.515 | 0.525 |
| | | MSE | 0.474 | 0.476 | 0.468 | 0.492 | 0.534 | 0.477 | 0.483 | 0.552 | 0.659 | 0.678 | 0.465 | 0.477 | 0.473 | 0.510 | 0.536 |
| ETTm2 | 96 | MAE | 0.199 | 0.198 | 0.197 | 0.207 | 0.202 | 0.199 | 0.198 | 0.201 | 0.238 | 0.210 | 0.199 | 0.198 | 0.197 | 0.206 | 0.197 |
| | | MSE | 0.078 | 0.078 | 0.077 | 0.082 | 0.080 | 0.079 | 0.077 | 0.079 | 0.105 | 0.086 | 0.078 | 0.078 | 0.077 | 0.083 | 0.077 |
| | 168 | MAE | 0.219 | 0.219 | 0.217 | 0.222 | 0.224 | 0.220 | 0.219 | 0.221 | 0.261 | 0.235 | 0.219 | 0.218 | 0.217 | 0.227 | 0.220 |
| | | MSE | 0.093 | 0.093 | 0.092 | 0.094 | 0.097 | 0.093 | 0.092 | 0.094 | 0.133 | 0.105 | 0.093 | 0.093 | 0.093 | 0.099 | 0.094 |
| | 336 | MAE | 0.242 | 0.241 | 0.242 | 0.246 | 0.250 | 0.245 | 0.245 | 0.249 | 0.302 | 0.275 | 0.242 | 0.241 | 0.240 | 0.258 | 0.250 |
| | | MSE | 0.113 | 0.113 | 0.114 | 0.114 | 0.121 | 0.114 | 0.114 | 0.120 | 0.169 | 0.142 | 0.113 | 0.113 | 0.113 | 0.126 | 0.122 |
| | 720 | MAE | 0.264 | 0.264 | 0.268 | 0.274 | 0.277 | 0.277 | 0.287 | 0.293 | 0.336 | 0.314 | 0.264 | 0.264 | 0.262 | 0.303 | 0.277 |
| | | MSE | 0.139 | 0.139 | 0.142 | 0.144 | 0.155 | 0.147 | 0.154 | 0.162 | 0.207 | 0.186 | 0.139 | 0.139 | 0.137 | 0.181 | 0.155 |
| Electricity | 96 | MAE | 0.265 | 0.266 | 0.284 | 0.278 | 0.273 | 0.244 | 0.248 | 0.280 | 0.303 | 0.275 | 0.254 | 0.258 | 0.269 | 0.289 | 0.251 |
| | | MSE | 0.181 | 0.181 | 0.189 | 0.189 | 0.198 | 0.148 | 0.152 | 0.171 | 0.195 | 0.172 | 0.159 | 0.165 | 0.164 | 0.185 | 0.151 |
| | 168 | MAE | 0.265 | 0.267 | 0.281 | 0.273 | 0.267 | 0.254 | 0.252 | 0.288 | 0.320 | 0.279 | 0.256 | 0.258 | 0.272 | 0.301 | 0.254 |
| | | MSE | 0.176 | 0.177 | 0.183 | 0.181 | 0.184 | 0.159 | 0.155 | 0.178 | 0.211 | 0.177 | 0.160 | 0.163 | 0.168 | 0.200 | 0.155 |
| | 336 | MAE | 0.285 | 0.288 | 0.301 | 0.296 | 0.289 | 0.270 | 0.272 | 0.312 | 0.335 | 0.299 | 0.272 | 0.278 | 0.287 | 0.312 | 0.266 |
| | | MSE | 0.190 | 0.191 | 0.198 | 0.197 | 0.201 | 0.166 | 0.167 | 0.197 | 0.222 | 0.192 | 0.169 | 0.175 | 0.177 | 0.207 | 0.162 |
| | 720 | MAE | 0.320 | 0.322 | 0.333 | 0.340 | 0.329 | 0.302 | 0.304 | 0.332 | 0.357 | 0.326 | 0.303 | 0.312 | 0.307 | 0.336 | 0.296 |
| | | MSE | 0.223 | 0.224 | 0.231 | 0.239 | 0.244 | 0.191 | 0.194 | 0.217 | 0.249 | 0.218 | 0.194 | 0.204 | 0.193 | 0.237 | 0.188 |
| Exchange | 96 | MAE | 0.167 | 0.167 | 0.166 | 0.202 | 0.164 | 0.168 | 0.182 | 0.168 | 0.278 | 0.223 | 0.167 | 0.169 | 0.166 | 0.220 | 0.170 |
| | | MSE | 0.053 | 0.053 | 0.054 | 0.070 | 0.053 | 0.055 | 0.061 | 0.055 | 0.183 | 0.096 | 0.053 | 0.054 | 0.054 | 0.087 | 0.057 |
| | 168 | MAE | 0.215 | 0.217 | 0.213 | 0.277 | 0.216 | 0.217 | 0.239 | 0.238 | 0.364 | 0.295 | 0.215 | 0.220 | 0.213 | 0.303 | 0.218 |
| | | MSE | 0.087 | 0.088 | 0.087 | 0.127 | 0.088 | 0.089 | 0.105 | 0.110 | 0.279 | 0.157 | 0.087 | 0.092 | 0.087 | 0.186 | 0.089 |
| | 336 | MAE | 0.291 | 0.297 | 0.304 | 0.332 | 0.312 | 0.295 | 0.329 | 0.406 | 0.566 | 0.375 | 0.290 | 0.303 | 0.305 | 0.439 | 0.314 |
| | | MSE | 0.158 | 0.162 | 0.171 | 0.190 | 0.178 | 0.164 | 0.184 | 0.305 | 0.603 | 0.252 | 0.156 | 0.165 | 0.171 | 0.318 | 0.183 |
| | 720 | MAE | 0.398 | 0.406 | 0.466 | 0.628 | 0.526 | 0.431 | 0.431 | 0.599 | 0.730 | 0.503 | 0.427 | 0.437 | 0.474 | 0.583 | 0.496 |
| | | MSE | 0.283 | 0.292 | 0.375 | 0.674 | 0.440 | 0.350 | 0.322 | 0.591 | 0.822 | 0.448 | 0.312 | 0.338 | 0.386 | 0.534 | 0.403 |
| Traffic | 96 | MAE | 0.334 | 0.334 | 0.374 | 0.403 | 0.556 | 0.323 | 0.314 | 0.323 | 0.351 | 0.372 | 0.325 | 0.340 | 0.358 | 0.391 | 0.371 |
| | | MSE | 0.403 | 0.403 | 0.443 | 0.513 | 0.738 | 0.386 | 0.364 | 0.365 | 0.415 | 0.455 | 0.393 | 0.393 | 0.409 | 0.458 | 0.434 |
| | 168 | MAE | 0.333 | 0.334 | 0.517 | 0.585 | 0.598 | 0.320 | 0.319 | 0.340 | 0.355 | 0.506 | 0.328 | 0.346 | 0.348 | 0.392 | 0.356 |
| | | MSE | 0.413 | 0.414 | 0.654 | 0.796 | 0.803 | 0.393 | 0.383 | 0.400 | 0.423 | 0.746 | 0.408 | 0.403 | 0.412 | 0.468 | 0.418 |
| | 336 | MAE | 0.345 | 0.346 | 0.371 | 0.394 | 0.379 | 0.336 | 0.333 | 0.403 | 0.376 | 0.636 | 0.345 | 0.357 | 0.356 | 0.403 | 0.366 |
| | | MSE | 0.436 | 0.437 | 0.463 | 0.511 | 0.520 | 0.425 | 0.406 | 0.518 | 0.459 | 1.048 | 0.436 | 0.426 | 0.437 | 0.498 | 0.444 |
| | 720 | MAE | 0.368 | 0.372 | 0.395 | 0.420 | 0.403 | 0.356 | 0.397 | 0.563 | 0.402 | 0.786 | 0.368 | 0.377 | 0.375 | 0.423 | 0.382 |
| | | MSE | 0.469 | 0.472 | 0.497 | 0.541 | 0.548 | 0.448 | 0.482 | 0.778 | 0.489 | 1.327 | 0.469 | 0.454 | 0.465 | 0.533 | 0.473 |
| Weather | 96 | MAE | 0.220 | 0.214 | 0.228 | 0.247 | 0.216 | 0.210 | 0.217 | 0.219 | 0.251 | 0.203 | 0.211 | 0.215 | 0.219 | 0.234 | 0.196 |
| | | MSE | 0.181 | 0.173 | 0.175 | 0.190 | 0.195 | 0.172 | 0.172 | 0.170 | 0.190 | 0.173 | 0.170 | 0.170 | 0.164 | 0.175 | 0.164 |
| | 168 | MAE | 0.259 | 0.254 | 0.258 | 0.285 | 0.242 | 0.250 | 0.247 | 0.253 | 0.303 | 0.248 | 0.252 | 0.253 | 0.257 | 0.270 | 0.232 |
| | | MSE | 0.218 | 0.210 | 0.206 | 0.226 | 0.231 | 0.211 | 0.208 | 0.206 | 0.255 | 0.228 | 0.209 | 0.206 | 0.203 | 0.213 | 0.207 |
| | 336 | MAE | 0.298 | 0.297 | 0.312 | 0.342 | 0.290 | 0.301 | 0.315 | 0.316 | 0.376 | 0.306 | 0.293 | 0.299 | 0.309 | 0.314 | 0.288 |
| | | MSE | 0.278 | 0.274 | 0.277 | 0.293 | 0.301 | 0.270 | 0.287 | 0.279 | 0.364 | 0.314 | 0.271 | 0.268 | 0.269 | 0.275 | 0.285 |
| | 720 | MAE | 0.346 | 0.345 | 0.358 | 0.400 | 0.327 | 0.366 | 0.368 | 0.379 | 0.454 | 0.350 | 0.331 | 0.340 | 0.355 | 0.355 | 0.356 |
| | | MSE | 0.343 | 0.339 | 0.338 | 0.366 | 0.359 | 0.349 | 0.360 | 0.368 | 0.479 | 0.386 | 0.329 | 0.322 | 0.331 | 0.336 | 0.348 |
| Count (1st) | | | 34 | 21 | 12 | 0 | 5 | 41 | 22 | 4 | 0 | 2 | 29 | 14 | 18 | 0 | 14 |

Table 12: Full results of the model parameter experiments on lookback and horizon lengths compared with FAN. The best results are highlighted in **bold**.

| Parameters | Lookback (L) | | | | | | Horizon (H) | | | | | |
|---|---|---|---|---|---|---|---|---|---|---|---|---|
| Lengths | 48 | 72 | 96 | 120 | 144 | 168 | 270 | 336 | 420 | 540 | 600 | 720 |
| MAE (Dual-AN) | **0.174** | **0.171** | **0.168** | **0.167** | **0.170** | **0.170** | **0.271** | **0.290** | **0.319** | **0.353** | **0.330** | **0.427** |
| MAE (FAN) | 0.194 | 0.193 | 0.182 | 0.179 | 0.189 | 0.178 | 0.280 | 0.303 | 0.324 | 0.360 | 0.355 | 0.437 |
| MSE (Dual-AN) | **0.057** | **0.056** | **0.055** | **0.055** | **0.055** | **0.055** | **0.134** | **0.156** | **0.191** | **0.219** | **0.202** | **0.312** |
| MSE (FAN) | 0.073 | 0.072 | 0.061 | 0.060 | 0.063 | 0.060 | 0.142 | 0.165 | 0.196 | 0.228 | 0.216 | 0.338 |

### D.3 MODEL EFFICIENCY

In terms of model efficiency, we compared the prediction performance, number of parameters, and training speed of Dual-AN and other normalization methods on the Traffic dataset on the Informer backbone with a prediction length of H = 720. The results are shown in Figure 5. With the training speed of Dual-AN no more than 3% different from that of other methods and the average number of parameters no more than 5%, the average MAE metric is improved by 33.71% and the average MSE metric is improved by 41.74%, which highlights the excellent performance of our Dual-AN model in balancing effect and efficiency. Although compared with the existing most advanced method, FAN, the reduction ratio can also reach 7.05% and 10.30%, which further demonstrates the superior performance and high efficiency of the Dual-AN method proposed in this paper.

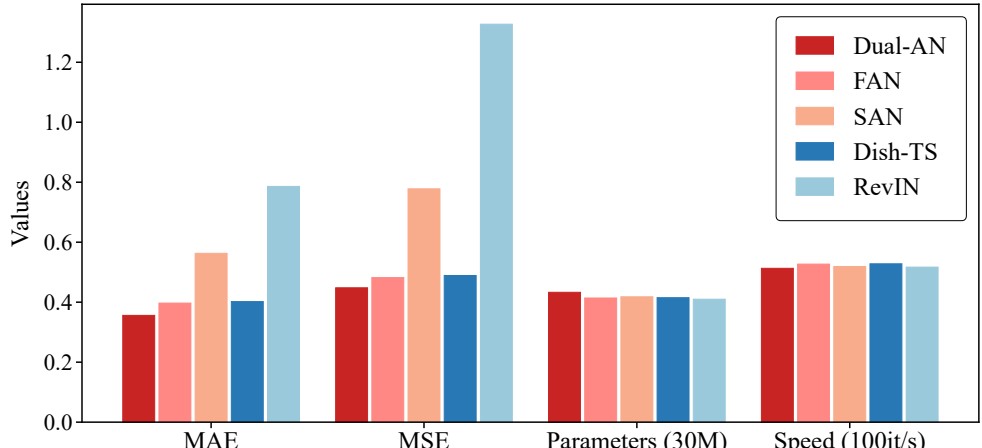

Figure 5: Model efficiency comparison of Dual-AN, FAN, SAN, Dish-TS, and RevIN.

To further illustrate the model efficiency of the Dual-AN method, we present a comparison of its training and testing times with other normalization methods in Table 13.

Table 13: The comparison of training time (single epoch) and testing time for 5 runs with fixed seeds of the pure backbone with and without the Dual-AN method and other normalization methods.

| Method | Training Time | Testing Time |
|---|---|---|
| Backbone | 96.4539±1.2890 | 13.6083±0.4669 |
| +Dual-AN | 119.6488±3.0048 | 14.6919±0.3710 |
| +FAN | 97.4063±1.8296 | 13.6021±0.6503 |
| +SAN | 101.7588±1.2157 | 15.1279±0.3996 |
| +Dish-TS | 98.5975±1.5698 | 13.7292±0.7178 |
| +RevIN | 97.2728±1.6602 | 13.7905±0.1818 |

The results clearly show that the training and inference times of Dual-AN are highly competitive with those of existing standardized baselines. Therefore, model complexity does not pose a practical concern and can be safely regarded as negligible in deployment.

## E  FORECAST SHOWCASES

To visualize the performance of our proposed Dual-AN method and since the FAN method Ye et al. (2024) is the most advanced among the existing methods, we illustrate the visual forecasting results of Dual-AN compared with FAN Ye et al. (2024) on the ETTh1 dataset with the Informer backbone in Figure 6.

As shown in Figure 6, in extreme cases of the time series (such as maximum and minimum values), Dual-AN can more accurately capture the local trends of the time series, demonstrating the

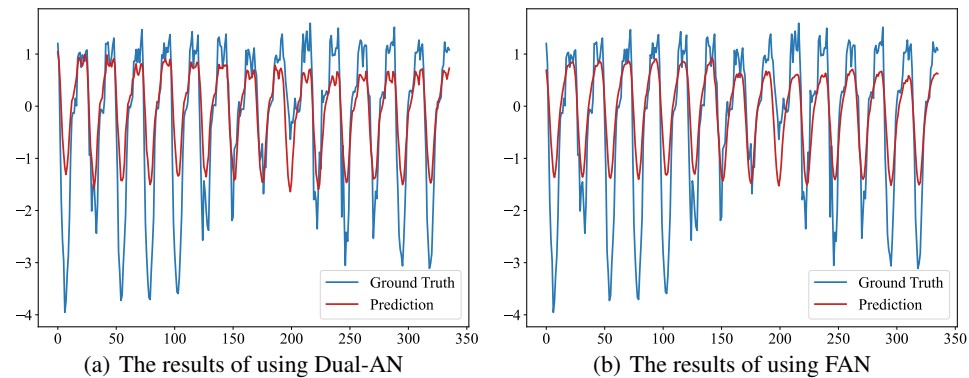

(a) The results of using Dual-AN        (b) The results of using FAN

Figure 6: The visual forecasting results of 336 steps of (a) Dual-AN and (b) FAN on the ETTh1 dataset with the Informer backbone.

significant advantages of the sliding window adaptive normalization (SWAN) module at a fine granularity, while furthermore making more accurate forecasting of future trends through the statistical prediction module (SPM).

In addition, we show the visual forecasting results of the baseline model (Informer) and the Dual-AN method proposed in this paper on the ETTh1 dataset in Figure 7, which once again corroborate the significant advantages of Dual-AN in capturing future local trends in both short-term and long-term forecasting.

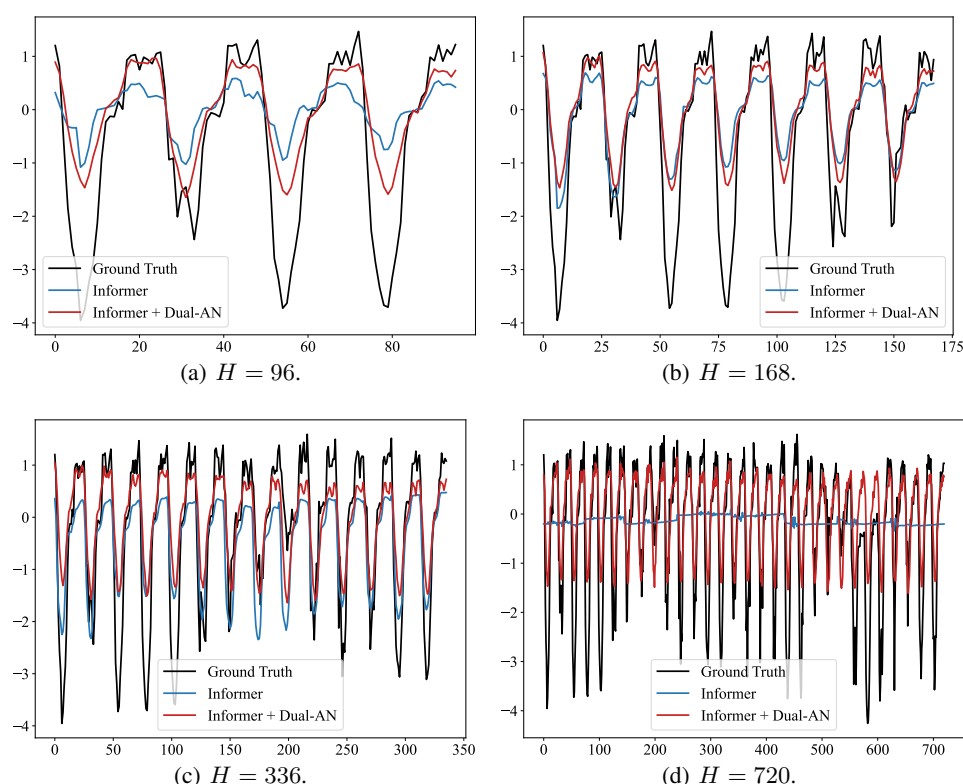

(a) $H = 96$.           (b) $H = 168$.

(c) $H = 336$.           (d) $H = 720$.

Figure 7: The visual forecasting results of backbone (Informer) and Dual-AN on the ETTh1 dataset across 4 different prediction lengths.

## F  FURTHER DISCUSSION ON THE WINDOW SIZE SELECTION PRINCIPLE

The criterion for dynamic window selection, which minimizes the standard deviation of local standard deviations (Equation 2), is rooted in the principle of seeking maximum statistical homogeneity at a given temporal scale. The underlying hypothesis is that an optimal normalization window should span a region where the series' intrinsic volatility is most stable. A stable volatility profile leads to more consistent scaling factors (mean and standard deviation), which in turn transforms the input into a sequence that more closely approximates a stationary process—a key assumption for many predictive models. While alternative criteria, such as those based on information theory (e.g., Minimum Description Length) or spectral entropy, could offer more theoretical grounding, the proposed heuristic provides a computationally efficient and empirically robust solution, as demonstrated by the analysis in Section 4.5.2. A rigorous theoretical exploration of optimal windowing strategies is a promising direction for future work.

To further illustrate the robustness of our dynamic window selection mechanism, we have expanded Table 5 in Table 14 to show the experimental results under a wider range of window size selection scenarios.

Table 14: The MAE and MSE experimental results of different window sizes. The best results are highlighted in **bold**.

| Window Size | Metrics | 96 | 168 | 336 | 720 | Count (1st) |
|---|---|---|---|---|---|---|
| 6 | MAE | 0.19876 | 0.21911 | 0.24262 | 0.26464 | 0 |
| | MSE | 0.07819 | 0.09329(0) | 0.11430 | 0.13932 | 0 |
| 12 | MAE | **0.19871** | **0.21893** | 0.24252 | 0.26448 | 2 |
| | MSE | 0.07813 | 0.09329(1) | 0.11431 | 0.13939 | 0 |
| 24 | MAE | 0.19887 | 0.21896 | 0.24286 | **0.26446** | 1 |
| | MSE | 0.07812 | **0.09325** | 0.11447 | 0.13940 | 1 |
| 36 | MAE | 0.19886 | 0.21947 | 0.24293 | 0.26466 | 0 |
| | MSE | 0.07823 | 0.09349 | 0.11454 | 0.13958 | 0 |
| 48 | MAE | 0.19884(4) | 0.21983 | **0.24153** | 0.2645 | 1 |
| | MSE | **0.07805** | 0.09367 | **0.11310** | **0.13929** | 3 |
| 60 | MAE | 0.19876 | 0.21916 | 0.24283 | 0.26467 | 0 |
| | MSE | 0.07823 | 0.09326 | 0.11460 | 0.13944 | 0 |
| 72 | MAE | 0.19883(7) | 0.21899 | 0.24275 | 0.26455 | 0 |
| | MSE | 0.07822 | 0.09327 | 0.11436 | 0.13934 | 0 |
| Mean±Std | MAE | 0.19881±0.00006 | 0.21921±0.00033 | 0.24258±00048 | 0.26457±0.00009 | - |
| | MSE | 0.07817±0.00007 | 0.09336±0.00016 | 0.11424±0.00052 | 0.13939±0.00010 | - |

As can be seen from Table 14, the standard deviations of MAE and MSE indices are all within 0.0005 under different window sizes across all 4 horizons, indicating that the differences between different window sizes are negligible.

### F.1  THEORETICAL JUSTIFICATION FOR WINDOW SIZE SELECTION

In this section, we provide a theoretical justification for the dynamic window size selection criterion used in our Dual-AN module. We demonstrate that selecting the window size $W$ to minimize the standard deviation of the sliding volatility estimates corresponds to optimizing the Bias-Variance trade-off under the assumption of local stationarity.

#### F.1.1  PROBLEM FORMULATION

Let the residual time series $r_t$ (after frequency decomposition) be modeled as a **Locally Stationary Process (LSP)**:

$$r_t = \sigma(t) \cdot \epsilon_t, \quad \epsilon_t \overset{i.i.d}{\sim} \mathcal{N}(0, 1) \tag{35}$$

where $\sigma(t)$ is a deterministic, slowly varying (or piecewise constant) volatility function, and $\epsilon_t$ represents stationary Gaussian noise.

Our goal is to estimate the local volatility $\sigma(t)$ using a sliding window estimator $\hat{\sigma}_{t,W}$ with window size $W$:

$$\hat{\sigma}_{t,W} = \sqrt{\frac{1}{W} \sum_{i=t-W+1}^{t} r_i^2} \tag{36}$$

The selection criterion proposed in the paper minimizes the temporal fluctuation of this estimator:

$$\mathcal{L}(W) = \text{StdDev}_t\left[\hat{\sigma}_{t,W}\right] = \sqrt{\frac{1}{T}\sum_{t=1}^{T}\left(\hat{\sigma}_{t,W} - \bar{\sigma}_W\right)^2} \tag{37}$$

### F.1.2 BIAS-VARIANCE TRADE-OFF ANALYSIS

The fluctuation metric $\mathcal{L}(W)$ is influenced by two competing sources of error: sampling variance (dominant at small $W$) and estimation bias (dominant at large $W$ near change points).

**Case 1: Small Window Size (Variance Domination)** Consider a locally stationary segment where the true volatility is constant, $\sigma(t) = \sigma_0$. For a window size $W$, the empirical variance $\hat{\sigma}_{t,W}^2$ follows a scaled Chi-squared distribution:

$$\hat{\sigma}_{t,W}^2 \sim \frac{\sigma_0^2}{W}\chi_W^2 \tag{38}$$

Using the standard approximation for the variance of the standard deviation estimator for Gaussian data, the variance of the estimator itself is inversely proportional to $W$:

$$\text{Var}\left(\hat{\sigma}_{t,W}\right) \approx \frac{\sigma_0^2}{2W} \tag{39}$$

**Implication:** When $W$ is small, the estimator $\hat{\sigma}_{t,W}$ is highly sensitive to the noise $\epsilon_t$. Even if the underlying $\sigma(t)$ is constant, the estimated sequence will fluctuate wildly solely due to sampling noise. This results in a high value of $\mathcal{L}(W)$. Increasing $W$ effectively suppresses this noise.

Consider a non-stationary transition where the volatility steps from $\sigma_1$ to $\sigma_2$ at time $\tau$. If $W$ is large relative to the local scale, the window will span across the change point for a long duration. During this transition, the estimator $\hat{\sigma}_{t,W}$ is a mixture of the two regimes. The expectation of the estimator becomes:

$$\mathbb{E}[\hat{\sigma}_{t,W}^2] \approx \alpha\sigma_1^2 + (1-\alpha)\sigma_2^2 \tag{40}$$

where $\alpha$ represents the proportion of the window in the first regime. **Implication:** An excessively large $W$ creates a "smearing" effect, introducing a bias that manifests as a slow, high-amplitude ramp in the estimator sequence as it slides across regimes. This structural variation contributes to the total fluctuation $\mathcal{L}(W)$. Furthermore, overly large windows fail to capture local adaptive characteristics, violating the local stationarity assumption.

### F.1.3 CONCLUSION

Our criterion $\min_W \mathcal{L}(W)$ effectively identifies the optimal scale by balancing these two factors:

1. It penalizes **undersized windows** where the signal is drowned out by the high variance of the estimator ($1/W$ term).

2. It penalizes **oversized windows** (implicitly) by favoring the scale where the estimator stabilizes within homogeneous segments without smoothing out necessary structural changes.

Thus, the selected window size represents the *characteristic scale of stationarity* for the given dataset, ensuring robust normalization.

## G DESIGN RATIONALE FOR THE STATISTICAL PREDICTION MODULE (SPM)

The selection of an MLP architecture for the SPM (Section 3.2) was a deliberate design choice balancing expressive power against computational cost. The SPM's task is to predict future window-level statistics—a sequence-to-sequence regression problem. Although more complex architectures

like RNNs or Transformers could be employed, they would introduce significant parameter overhead and computational latency. Crucially, the sequences of statistical moments (mean and standard deviation) are typically much smoother and less noisy than the raw time series data. Consequently, an MLP, as a universal function approximator, possesses sufficient expressive capacity to model these smoother dynamics effectively. This was confirmed during preliminary experiments, where replacing the MLP with an LSTM yielded only marginal performance gains at the cost of a substantial increase in training time, thus justifying the current, more efficient design. This ensures that Dual-AN remains a lightweight and broadly applicable plug-in.

## H    ABLATION STUDY ON LOSS FUNCTION COMPONENTS

To validate the effectiveness of the dual-component loss function described in Section 3.3, an additional ablation study was conducted. The full model, optimized with the combined loss ($\mathcal{L}_{nonstat} + \mathcal{L}_{stat}$), is compared against a variant trained with a single loss function applied only to the final prediction (i.e., MSE on the final output $\hat{Y}$). As shown in Table 15, explicitly supervising both the non-stationary and stationary components leads to improved forecasting accuracy. This result supports the hypothesis that the dual loss acts as a valuable regularizer, guiding the model toward a more meaningful and effective decomposition of the time series, which ultimately enhances prediction quality.

Table 15: Ablation study on loss function components on the ETTh1 dataset with the Informer backbone (H=336).

| Loss Configuration | MAE | MSE |
|---|---|---|
| Single Loss on Final Prediction ($\mathcal{L}(\hat{Y}, Y)$) | 0.501 | 0.462 |
| Dual Loss ($\mathcal{L}_{nonstat} + \mathcal{L}_{stat}$) | **0.493** | **0.452** |

## I    COMPUTATIONAL COMPLEXITY ANALYSIS

The computational overhead introduced by Dual-AN stems from the SWAN and SPM modules. Let $N$ be the number of variables, $L$ be the lookback length, and $W_{opt}$ be the optimal window size.

- **SWAN**: The primary cost is the calculation of sliding window statistics. A naive implementation has a time complexity of $\mathcal{O}(L \cdot W_{opt} \cdot N)$. However, this can be optimized to $\mathcal{O}(L \cdot N)$ using moving average algorithms. The space complexity is $\mathcal{O}(L \cdot N)$ to store the statistics for each time step.
- **SPM**: The complexity is determined by its MLP layers. For the structure described in Appendix C.2, the complexity is independent of the sequence length and depends only on the hidden dimensions, which are fixed hyperparameters. Thus, its complexity is $\mathcal{O}(N)$.

The total additional time complexity is therefore approximately $\mathcal{O}(L \cdot N)$. This is linear with respect to the input sequence length and does not alter the dominant complexity of most modern backbone models (e.g., $\mathcal{O}(L^2 \cdot N)$ for standard Transformers or $\mathcal{O}(L \cdot \log L \cdot N)$ for Informer). This analysis confirms that Dual-AN is a computationally feasible plug-in for a wide range of applications without introducing a new performance bottleneck.

## J    PRACTICAL GUIDANCE FOR CHOOSING NORMALIZATION METHODS

To make Dual-AN easier to apply in practice, we provide a simple rule-of-thumb on when to prefer Dual-AN over simpler normalization schemes such as RevIN. Following FAN (**?**), we characterize

Table 16: Practical guidance for choosing normalization methods based on dataset characteristics. TV and SV are computed as in FAN Ye et al. (2024). "Avg. gain vs RevIN" is the relative MAE reduction of Dual-AN compared with RevIN, averaged over three backbones and four horizons as shown in Table 2.

| Dataset | TV | SV | Variation level | Avg. MAE gain vs RevIN (%) | Recommended normalization |
|---------|------|--------|-----------------|----------------------------|---------------------------|
| ETTh1 | 3.839 | 3.690 | High | 9.6% | Dual-AN (recommended) |
| ETTh2 | 0.154 | 1.013 | Moderate | 8.9% | Dual-AN (recommended) |
| ETTm1 | 0.030 | 3.330 | High | 8.4% | Dual-AN (recommended) |
| ETTm2 | 0.196 | 1.648 | Moderate | 4.9% | Dual-AN / RevIN (both acceptable) |
| Electricity | 0.249 | 0.435 | Moderate | 3.5% | Dual-AN / RevIN (both acceptable) |
| Exchange | 0.242 | 2.645 | Moderate | 13.9% | Dual-AN (recommended) |
| Traffic | 0.068 | 14.225 | High | 28.5% | Dual-AN (strongly recommended) |
| Weather | 0.028 | 0.387 | Low | -2.6% | RevIN (near-stationary series) |

each dataset by:
- **Trend Variation (TV)**, which measures the distributional shift of the global trend across the train/validation/test splits;
- **Seasonality Variation (SV)**, which measures how much the spectral (seasonal) components change across these splits.

Larger TV or SV indicates stronger non-stationarity in the time or frequency domain, respectively. For each dataset, we further compute the average MAE improvement of Dual-AN over RevIN,

$$\Delta_{\text{MAE}} = \frac{\text{MAE}_{\text{RevIN}} - \text{MAE}_{\text{Dual-AN}}}{\text{MAE}_{\text{RevIN}}} \times 100\%,$$

averaged over the three backbones (DLinear, Informer, SCINet) and four prediction horizons using the results in Table 2.

Based on TV/SV, we divide series into three regimes:

- **Low variation**: TV $< 0.05$ *and* SV $< 0.5$ (close to stationary);
- **High variation**: TV $\geq 1.0$ *or* SV $\geq 3.0$ (strong trend/seasonality shifts);
- **Moderate variation**: all remaining cases.

Table 16 summarizes the statistics and our recommended normalization choice for each benchmark dataset. In short, Dual-AN is clearly preferred for moderate or high variation, while RevIN is slightly better on datasets that are nearly stationary in both trend and seasonality (e.g., Weather).

## K LIMITATIONS AND FUTURE WORK

Based on the comprehensive framework and experimental results presented in this paper, we identify several limitations and suggest promising avenues for future work. First, the current implementation of the sliding window adaptive normalization (SWAN) module relies on a pre-defined set of candidate window sizes, which may not be optimal for all types of time series. Although Dual-AN exhibits robustness across various window sizes, integrating an adaptive mechanism to dynamically determine window size during training could further improve model flexibility and generalization. Second, while Dual-AN achieves significant improvements across multiple backbones and datasets, its performance on series with extremely low trend and seasonality variations (e.g., Weather) remains less competitive compared to specialized methods like RevIN Kim et al. (2021). This suggests that a more nuanced integration of time and frequency domains may be necessary for such scenarios. Future work will focus on developing automated window size selection algorithms and designing backbone-specific variants of Dual-AN to enhance its applicability and performance. Addressing these aspects will further establish Dual-AN as a versatile and powerful framework for non-stationary time series forecasting.

