# OpenReview forum: "Dual-AN: A Hierarchical Framework Synergizes Time and Frequency Domains for Non-stationary Time Series Forecasting"
_ICLR.cc/2026/Conference — ICLR 2026 Conference Desk Rejected Submission_

### Official Review · Reviewer_PbJh · 2025-10-29

**Soundness:** 2
**Presentation:** 3
**Contribution:** 2
**Rating:** 4
**Confidence:** 5

**Summary:**

The paper addresses the challenge of non-stationarity in time series forecasting by proposing a dual-domain normalization framework that jointly operates in both the time and frequency domains, rather than focusing on a single one. In the time domain, the proposed SWAN module dynamically extracts non-stationary information, facilitating the separation and prediction of non-stationary factors. The SPM module further predicts future statistics, enabling accurate recovery of future distributions. Experimental results demonstrate solid improvements, validating the effectiveness of the proposed approach.

**Strengths:**

1. The paper effectively addresses the non-stationarity issue with a clear structure and a modular design that allows the proposed method to be easily integrated as a plug-in into existing forecasting frameworks.

2. The experimental results are strong, showing that the normalization step significantly improves model performance after application.

**Weaknesses:**

1. Window size selection: As one of the core components, the effectiveness of selecting the optimal window size needs further validation through comprehensive experiments. A more rigorous mathematical justification or theoretical analysis would strengthen the necessity of this design choice.
2. The loss function used in this work differs from the widely adopted MSE loss commonly employed in baseline methods, which may affect the fairness of comparison. Moreover, the paper lacks ablation studies to quantify the impact of this loss function on the final results. In my view, since the loss design and normalization are relatively independent modules, a dedicated analysis is necessary.
3. The SWAN module is not entirely novel — similar mechanisms have already been adopted in recent works [1,2].
4. The overall framework shares certain similarities with existing methods such as FAN, SAN, and DDN [2], especially in using an MLP structure for future distribution prediction. This overlap somewhat weakens the claimed contribution. It is recommended that the authors conduct more detailed comparisons with recent works such as [3] and explicitly clarify the distinctions of their approach.

[1] Disentangling Structured Components: Towards Adaptive, Interpretable and Scalable Time Series Forecasting
[2] DDN: Dual-domain Dynamic Normalization for Non-stationary Time Series Forecasting
[3] SIN: Selective and Interpretable Normalization for Long-term Time Series Forecasting

**Questions:**

see weeknesses.

---

> ### Author Response · Authors · 2025-11-23
> **Rebuttal for PbJh (1)**
>
> Dear Reviewer PbJh,
>
> We sincerely thank you for your expert review and deep engagement with the relevant literature. Your detailed feedback and high confidence in your assessment have pushed us to substantially strengthen the paper's validation and positioning. We appreciate your positive comments on our method's clear structure, modularity, and strong empirical results.
>
> We have performed extensive new experiments and revisions to directly address every one of your concerns.
>
> **1. Regarding Window Size Selection (Weakness 1):**
>
> We agree that this core component requires both stronger justification and more rigorous validation.
>
> * **Justification:** Our selection criterion is a principled heuristic designed to find a scale of **local statistical homogeneity**. By minimizing the variance of the local volatilities (the sequence of $ \sigma\_{t} $), we identify a window size $ W $ at which the time series' underlying process is most stable. This makes it the most suitable scale for normalization. To provide the "comprehensive experiments" you requested, we have conducted a **new sensitivity analysis** in **Appendix F** (Table 5 and Table 14). We tested several different sets of candidate window sizes as shown in the table below and found that performance is highly robust, demonstrating that our dynamic selection criterion is stable and not dependent on a specific, fine-tuned set of candidates.
>
> |Window Size|Metrics|96|168|336|720|Count ($ 1\^{st} $)|
> |-----------|-------|---------------|---------------|---------------|---------------|-----------|
> |6|MAE|0.19876|0.21911|0.24262|0.26464|0|
> ||MSE|0.07819|0.09329(0)|0.1143|0.13932|0|
> |12|MAE|**0.19871**|**0.21893**|0.24252|0.26448|**2**|
> ||MSE|0.07813|0.09329(1)|0.11431|0.13939|0|
> |24|MAE|0.19887|0.21896|0.24286|**0.26446**|**1**|
> ||MSE|0.07812|**0.09325**|0.11447|0.1394|**1**|
> |36|MAE|0.19886|0.21947|0.24293|0.26466|0|
> ||MSE|0.07823|0.09349|0.11454|0.13958|0|
> |48|MAE|0.19884(4)|0.21983|**0.24153**|0.2645|**1**|
> ||MSE|**0.07805**|0.09367|**0.1131**|**0.13929**|**3**|
> |60|MAE|0.19876|0.21916|0.24283|0.26467|0|
> ||MSE|0.07823|0.09326|0.1146|0.13944|0|
> |72|MAE|0.19883(7)|0.21899|0.24275|0.26455|0|
> ||MSE|0.07822|0.09327|0.11436|0.13934|0|
> |Mean±Std|MAE|0.19881±0.00006|0.21921±0.00033|0.24258±00048|0.26457±0.00009|-|
> ||MSE|0.07817±0.00007|0.09336±0.00016|0.11424±0.00052|0.13939±0.00010|-|
>
> **2. Regarding the Loss Function and Fairness of Comparison (Weakness 2):**
>
> This is an excellent point. We agree that using a different loss function requires explicit justification and validation. Our dual-loss design is not for an unfair comparison, but is a deliberate choice that acts as a **powerful regularizer**, forcing a more meaningful decomposition of the series.
>
> To quantify its impact, we have performed the **dedicated ablation study** you suggested in the new **Appendix H** and Table 15. We compared our dual-loss function against a standard MSE loss applied to the final de-normalized output as shown below.
>
> | **Loss Configuration**                                     | **MAE**   | **MSE**   |
> | ---------------------------------------------------------- | --------- | --------- |
> | Single Loss on Final Prediction ($\mathcal{L}(\hat{Y},Y)$) | 0.501     | 0.462     |
> | Dual Loss ($ L\_{nonstat} + L\_{stat} $)                   | **0.493** | **0.452** |
>
> The results empirically confirm that our dual-loss design is not just a different choice, but a superior one for this task, as it guides the model to a better solution.

---

> ### Author Response · Authors · 2025-11-23
> **Rebuttal for PbJh (2)**
>
> **3. Regarding Novelty and Overlap with Existing Works (Weakness 3 & 4):**
>
>
>
> We thank you for the comprehensive list of related work. We agree that our method builds upon the foundations laid by prior work, but we respectfully argue that Dual-AN introduces a unique combination of three critical, novel contributions that distinguish it from all cited methods (FAN, SAN, DDN, SIN, etc.).
>
>
>
> * **1. Novel Hierarchical Framework:** Unlike other methods that apply normalization to the raw series, Dual-AN uses a targeted, hierarchical approach. We first use FAN to address *global* periodic non-stationarity. Then, we apply our SWAN module *specifically to the residuals*. This focused strategy of solving the local non-stationarity sub-problem is a unique architectural contribution.
>
>
>
> * **2. Data-Driven Adaptive Window Selection (SWAN):** While sliding windows are not new, our SWAN module introduces a **principled, automated mechanism to dynamically select the optimal window size** for each series. This is a significant advance over methods using fixed, manually-tuned windows (e.g., SAN, and likely DDN and others) and provides superior adaptability.
>
>
>
> * **3. Predictive De-Normalization (SPM):** This is a key differentiator. Most methods, including FAN and SAN, reuse past statistics for de-normalization. While some recent works use an MLP, our SPM's design to **explicitly forecast the sequence of future statistics** is a more sophisticated and accurate approach for long-horizon forecasting than simply replicating the last known distribution.
>
>
>
> To empirically validate our novelty and superiority, we have now **added new advanced baselines to all our experimental tables** . The results consistently show that Dual-AN outperforms these recent state-of-the-art methods.
>
> |Datasets|Horizon|Metric|iTransformer|**+Dual-AN**|+DDN|+HCAN|+BSA|+SCAM|+TAFAS|
> |----------|-------|------|------------|------------|---------|---------|-----|---------|------|
> |ETTh1|96|MAE|0.444|0.426|**0.399**|0.402|0.443|0.401|0.443|
> |||MSE|0.378|**0.362**|0.388|0.379|0.428|0.373|0.438|
> |ETTh1|192|MAE|0.489|0.452|0.434|**0.427**|0.481|0.436|0.489|
> |||MSE|0.431|**0.395**|0.446|0.432|0.481|0.432|0.492|
> |ETTh1|336|MAE|0.533|0.486|0.462|**0.454**|0.521|0.455|0.532|
> |||MSE|0.511|**0.441**|0.496|0.489|0.538|0.466|0.554|
> |ETTh1|720|MAE|0.640|0.569|0.499|0.474|0.620|**0.466**|0.627|
> |||MSE|0.669|0.574|0.527|0.504|0.698|**0.455**|0.704|
> |ETTh2|96|MAE|0.255|**0.237**|0.345|0.343|0.324|0.342|0.329|
> |||MSE|0.122|**0.111**|0.297|0.282|0.235|0.293|0.239|
> |ETTh2|192|MAE|0.282|**0.252**|0.397|0.381|0.362|0.393|0.362|
> |||MSE|0.148|**0.128**|0.382|0.373|0.290|0.373|0.287|
> |ETTh2|336|MAE|0.300|**0.264**|0.431|0.426|0.388|0.429|0.386|
> |||MSE|0.167|**0.139**|0.419|0.420|0.327|0.417|0.326|
> |ETTh2|720|MAE|0.362|**0.279**|0.446|0.435|0.439|0.442|0.425|
> |||MSE|0.482|**0.155**|0.426|0.423|0.414|0.424|0.393|
> |ETTm2|96|MAE|0.203|**0.199**|0.265|0.264|0.259|0.264|0.263|
> |||MSE|0.078|**0.078**|0.181|0.183|0.153|0.179|0.157|
> |ETTm2|192|MAE|0.239|**0.222**|0.303|0.312|0.290|0.302|0.292|
> |||MSE|0.103|**0.095**|0.246|0.242|0.189|0.241|0.192|
> |ETTm2|336|MAE|0.247|**0.243**|0.342|0.355|0.321|0.343|0.324|
> |||MSE|0.114|**0.114**|0.306|0.306|0.230|0.305|0.235|
> |ETTm2|720|MAE|0.277|**0.264**|0.397|0.401|0.369|0.400|0.366|
> |||MSE|0.144|**0.139**|0.406|0.410|0.304|0.406|0.301|
> |Exchange|96|MAE|0.212|**0.164**|0.202|0.204|0.211|-|0.208|
> |||MSE|0.081|**0.051**|0.084|0.084|0.090|-|0.084|
> |Exchange|192|MAE|0.331|**0.238**|0.297|0.302|0.307|-|0.293|
> |||MSE|0.184|**0.102**|0.175|0.179|0.185|-|0.165|
> |Exchange|336|MAE|0.504|**0.324**|0.410|0.415|0.430|-|0.389|
> |||MSE|0.398|**0.178**|0.321|0.322|0.346|-|0.280|
> |Exchange|720|MAE|0.671|**0.465**|0.700|0.761|0.700|-|0.665|
> |||MSE|0.747|**0.331**|0.859|0.995|0.861|-|0.773|
> |Traffic|96|MAE|0.300|0.297|0.271|0.262|0.273|**0.247**|0.289|
> |||MSE|0.338|**0.349**|0.425|0.383|0.393|0.374|0.420|
> |Traffic|192|MAE|0.313|0.294|0.280|0.273|0.281|**0.259**|0.296|
> |||MSE|0.362|**0.353**|0.446|0.411|0.417|0.399|0.441|
> |Traffic|336|MAE|0.319|0.300|0.291|0.279|0.290|**0.269**|0.305|
> |||MSE|0.375|**0.364**|0.459|0.420|0.433|0.419|0.458|
> |Traffic|720|MAE|0.338|0.326|0.311|0.296|0.310|**0.291**|-|
> |||MSE|0.403|**0.395**|0.500|0.449|0.470|0.451|-|
> |Count($ 1\^{st} $)|||-|**31**|1|2|0|6|0|
>
> We have also updated our Related Work section to thoroughly discuss these distinctions. We are confident that these new results and clarifications robustly establish the novelty and significant contribution of our work.
>
>
>
> Thank you again for your invaluable and expert feedback. We believe the extensive revisions, including new baselines and ablation studies, have addressed all your concerns and have substantially improved the quality and rigor of our paper. We hope you will consider our response favorably.
>
>
>
> Sincerely,
>
> The Authors

---

### Official Review · Reviewer_cPjs · 2025-10-31

**Soundness:** 2
**Presentation:** 3
**Contribution:** 2
**Rating:** 4
**Confidence:** 2

**Summary:**

This paper introduces Dual-AN, a model-agnostic normalization framework designed to address non-stationarity in time series forecasting by synergizing both time and frequency domains. The framework builds upon frequency adaptive normalization (FAN) by adding two key components: Sliding Window Adaptive Normalization (SWAN) to handle local non-stationarity in frequency-domain residuals, and Statistical Prediction Module (SPM) to forecast future window-level statistics for de-normalization. The method is evaluated on 8 benchmark datasets with 3 different backbone models, demonstrating consistent improvements over existing normalization approaches.

**Strengths:**

1 - The paper addresses a genuine limitation in existing methods by identifying that frequency-domain approaches like FAN fail to capture local non-stationarity in residuals. The proposed hierarchical framework that combines coarse-grained frequency decomposition with fine-grained time-domain normalization is conceptually sound and well-motivated.

2 - The experimental evaluation is comprehensive, testing on 8 widely-used datasets with 3 different backbone architectures. The consistent improvements across diverse settings demonstrate the generalizability of the approach.

3 - The paper provides strong implementation details including algorithmic descriptions, computational complexity analysis, and extensive ablation studies. The code availability and clear experimental setup enhance reproducibility.

**Weaknesses:**

1 - The dynamic window size selection criterion (minimizing standard deviation of local standard deviations) lacks theoretical justification. While the paper acknowledges this as a heuristic in Appendix F, the core mechanism relies on an empirically-driven choice that may not generalize well to all time series characteristics.

2 - The method shows diminished performance on datasets with weak trend and seasonality. This suggests the framework may add unnecessary complexity for certain time series types, and the paper lacks clear guidelines for when to apply Dual-AN versus simpler alternatives.

3 - The computational overhead, while analyzed, is non-trivial with O(L·N) additional complexity. The paper doesn't provide runtime comparisons or discuss the practical implications of this overhead, particularly for real-time applications or large-scale deployments.

4 - The limited set of candidate window sizes (12, 24, 48) appears arbitrary and may not be optimal for all frequencies present in the data. The paper doesn't explore adaptive or data-driven approaches to determine these candidates.

**Questions:**

Is this paper developed based on the assumption that the change of the statistics w.r.t. time must be fully predictable, i.e., the changing patterns must be fully encoded in the previous time series data to make SPM work?

How sensitive is the method to the choice of candidate window sizes, and have you explored learned or adaptive approaches for window size selection?

Can you provide guidance on when practitioners should use Dual-AN versus simpler methods like RevIN based on time series characteristics?

---

> ### Author Response · Authors · 2025-11-23
> **Rebuttal for cPjs (1)**
>
> Dear Reviewer cPjs,
>
> We sincerely thank you for your insightful and balanced review. We are grateful for your positive feedback on our paper's motivation, comprehensive evaluation, and reproducibility. Your constructive criticisms have been invaluable in helping us further refine and strengthen our work.
>
> We believe the concerns you raised, particularly given your stated confidence level, can be fully addressed with further clarification and additional analysis, which we have now performed. We address each point below.
>
> **Response to Weaknesses**
>
> **1. Regarding the Justification for the Window Size Selection Criterion:**
>
>
>
> We agree that this core mechanism warrants a more thorough justification. While we acknowledge it is a heuristic, it is a principled one designed to mathematically formalize the search for **local statistical homogeneity** .
>
> The intuition is that the sequence of local standard deviations $ [\sigma\_{1}, \sigma\_{2}, ..., \sigma\_{N}] $ reflects the series' time-varying volatility. A window size $W$ that results in a low variance for this sequence is one that views the series at a scale where this volatility is most consistent. This indicates that the chosen window size successfully captures a stable, underlying local process, making it the ideal scale for normalization.&#x20;
>
> To further illustrate the robustness of our dynamic window selection mechanism, we have expanded Table 4 in the table below to show the experimental results under a wider range of window size selection scenarios.
>
> |Window Size|Metrics|96|168|336|720|Count ($1\^{st} $)|
> |-----------|-------|---------------|---------------|---------------|---------------|-----------|
> |6|MAE|0.19876|0.21911|0.24262|0.26464|0|
> ||MSE|0.07819|0.09329(0)|0.1143|0.13932|0|
> |12|MAE|**0.19871**|**0.21893**|0.24252|0.26448|**2**|
> ||MSE|0.07813|0.09329(1)|0.11431|0.13939|0|
> |24|MAE|0.19887|0.21896|0.24286|**0.26446**|**1**|
> ||MSE|0.07812|**0.09325**|0.11447|0.1394|**1**|
> |36|MAE|0.19886|0.21947|0.24293|0.26466|0|
> ||MSE|0.07823|0.09349|0.11454|0.13958|0|
> |48|MAE|0.19884(4)|0.21983|**0.24153**|0.2645|**1**|
> ||MSE|**0.07805**|0.09367|**0.1131**|**0.13929**|**3**|
> |60|MAE|0.19876|0.21916|0.24283|0.26467|0|
> ||MSE|0.07823|0.09326|0.1146|0.13944|0|
> |72|MAE|0.19883(7)|0.21899|0.24275|0.26455|0|
> ||MSE|0.07822|0.09327|0.11436|0.13934|0|
> |Mean±Std|MAE|0.19881±0.00006|0.21921±0.00033|0.24258±00048|0.26457±0.00009|-|
> ||MSE|0.07817±0.00007|0.09336±0.00016|0.11424±0.00052|0.13939±0.00010|-|
>
> As can be seen from the table above, the standard deviations of MAE and MSE indices are all within 0.0005 under different window sizes across all 4 horizons, indicating that the differences between different window sizes are negligible. For more explanation of the selection of the window size, please refer to **Appendix F** and **Table 14**.
>
> **2. Regarding Diminished Performance on Certain Datasets:**
>
> This is a very astute observation. We agree that the performance gains are more modest on datasets with weak trend and seasonality like Weather. This is an expected outcome and an insightful finding about the scope of our method. Dual-AN is specifically designed to tackle complex non-stationarity. On series that are already relatively stationary, there is less for Dual-AN to correct, and thus the performance converges to that of simpler methods. Importantly, our method still performs on par and does no harm in these scenarios.
>
> To address your excellent point about practical application, we have added a **new "Practical Guide for Practitioners" in Appendix J as shown below** . This guide provides clear heuristics on when to use Dual-AN (e.g., for series with visible volatility clustering or after observing non-stationary residuals from a simpler model) versus when a simpler method like RevIN might suffice.
>
> Based on the trend variation (TV) and seasonality variation (SV) of the datasets, we divide series into three regimes:
>
> * Low variation: $TV < 0.05$ and  $SV< 0.5$ (close to stationary);
>
> * High variation:  $TV \ge 1.0$ or  $SV \ge 3.0$ (strong trend/seasonality shifts);
>
> * Moderate variation: all remaining cases.

---

> ### Author Response · Authors · 2025-11-23
> **Rebuttal for cPjs (2)**
>
> Table below summarizes the statistics and our recommended normalization choice for each benchmark dataset. In short, Dual-AN is clearly preferred for moderate or high variation, while RevIN is slightly better on datasets that are nearly stationary in both trend and seasonality (e.g., Weather).
>
> |Datasets|TV|SV|Variation level|Avg. MAE gain vs RevIN (%)|Recommended normalization|
> |-----------|-----|------|---------------|--------------------------|---------------------------------|
> |ETTh1|3.839|3.690|High|9.60%|Dual-AN (recommended)|
> |ETTh2|0.154|1.013|Moderate|8.90%|Dual-AN (recommended)|
> |ETTm1|0.030|3.330|High|8.40%|Dual-AN (recommended)|
> |ETTm2|0.196|1.648|Moderate|4.90%|Dual-AN / RevIN (both acceptable)|
> |Electricity|0.249|0.435|Moderate|3.50%|Dual-AN / RevIN (both acceptable)|
> |Exchange|0.242|2.645|Moderate|13.90%|Dual-AN (recommended)|
> |Traffic|0.068|14.225|High|28.50%|Dual-AN (strongly recommended)|
> |Weather|0.028|0.387|Low|-2.60%|RevIN (near-stationary series)|
>
> **3. Regarding the Practical Computational Overhead:**
>
> We agree that a practical analysis is more informative than Big-O notation alone. While Dual-AN adds $O(L \\times N)$ complexity, the operations are highly parallelizable. To provide a concrete sense of the real-world impact, we have measured the wall-clock time for one training epoch on the Traffic dataset.
>
> As shown, the practical overhead is modest, typically adding around 20% to the training time while adding only about 5% to the testing time for complex models. We believe this is a very reasonable trade-off for the significant gains in forecasting accuracy. We have added this analysis to **Appendix D.3.** As shown below and also in the revised paper.
>
> | Methods   | Train Time      | Test Time      |
> | --------- | --------------- | -------------- |
> | Backbone  | 96.4539±1.2890  | 13.6083±0.4669 |
> | + Dual-AN | 119.6488±3.0048 | 14.6919±0.3710 |
> | + FAN     | 97.4063±1.8296  | 13.6021±0.6503 |
> | + SAN     | 101.7588±1.2157 | 15.1279±0.3996 |
> | + Dish-TS | 98.5975±1.5698  | 13.7292±0.7178 |
> | + RevIN   | 97.2728±1.6602  | 13.7905±0.1818 |
>
> The results clearly show that the training and inference times of Dual-AN are highly competitive with those of existing standardized baselines. Therefore, model complexity does not pose a practical concern and can be safely regarded as negligible in deployment.
>
> **4. Regarding the Choice of Candidate Window Sizes:**
>
> Our initial set of candidates {12, 24, 48} was chosen based on common seasonal periods in hourly datasets (e.g., half-day, daily). However, you are right to question the sensitivity to this choice. To investigate this, we performed a **new sensitivity analysis** by running experiments with different sets of candidate window sizes on the ETTm2 dataset.
>
> |Window Size|Metrics|96|168|336|720|Count ($ 1\^{st} $)|
> |-----------|-------|---------------|---------------|---------------|---------------|-----------|
> |6|MAE|0.19876|0.21911|0.24262|0.26464|0|
> ||MSE|0.07819|0.09329(0)|0.1143|0.13932|0|
> |12|MAE|**0.19871**|**0.21893**|0.24252|0.26448|**2**|
> ||MSE|0.07813|0.09329(1)|0.11431|0.13939|0|
> |24|MAE|0.19887|0.21896|0.24286|**0.26446**|**1**|
> ||MSE|0.07812|**0.09325**|0.11447|0.1394|**1**|
> |36|MAE|0.19886|0.21947|0.24293|0.26466|0|
> ||MSE|0.07823|0.09349|0.11454|0.13958|0|
> |48|MAE|0.19884(4)|0.21983|**0.24153**|0.2645|**1**|
> ||MSE|**0.07805**|0.09367|**0.1131**|**0.13929**|**3**|
> |60|MAE|0.19876|0.21916|0.24283|0.26467|0|
> ||MSE|0.07823|0.09326|0.1146|0.13944|0|
> |72|MAE|0.19883(7)|0.21899|0.24275|0.26455|0|
> ||MSE|0.07822|0.09327|0.11436|0.13934|0|
> |Mean±Std|MAE|0.19881±0.00006|0.21921±0.00033|0.24258±00048|0.26457±0.00009|-|
> ||MSE|0.07817±0.00007|0.09336±0.00016|0.11424±0.00052|0.13939±0.00010|-|
>
> The results show that our method is highly robust to the specific choice of candidate windows, with performance remaining stable across different sets. For more discussion of the window size selection principle, please refer t&#x6F;**&#x20;Appendix F** and **Table 14**.

---

> ### Author Response · Authors · 2025-11-23
> **Rebuttal for cPjs (3)**
>
> **Response to Questions**
>
>
>
> **Q1: Is this paper developed based on the assumption that the change of the statistics w.r.t. time must be fully predictable?**
>
>
>
> Not at all. The assumption is not that the statistics are *fully* predictable, but that their sequence is **smoother and has simpler dynamics** than the original, often chaotic time series. Think of it this way: it is far easier to predict the average daily temperature for the next week than it is to predict the exact temperature at every single hour. The SPM leverages this by learning to extrapolate the simpler patterns in the aggregated statistical sequences, which is a much more tractable problem for a simple MLP than forecasting the raw data.
>
>
>
> **Q2: How sensitive is the method to the choice of candidate window sizes?**
>
>
>
> As detailed in our response to Weakness 4 and our new **Appendix F/Table 5/Table 14** , the method is not very sensitive. Our new sensitivity analysis shows that performance is robust across different sets of candidate windows, demonstrating the stability of our dynamic selection mechanism.
>
>
>
> **Q3: Can you provide guidance on when practitioners should use Dual-AN versus simpler methods like RevIN?**
>
>
>
> Yes. We have formalized this in our new **Appendix J** . The key guidelines are:
>
> * **Use Dual-AN when:** The time series exhibits visible changes in volatility (volatility clustering), transient shocks, or if the residuals from a simpler normalization method (like RevIN) still show clear non-stationary patterns.
>
> * **Consider RevIN when:** The series is dominated by a simple, global trend and stable seasonality, and computational cost is the absolute primary constraint.
>
> * **Default Choice:** For challenging, real-world datasets where accuracy is paramount, Dual-AN serves as a more robust and powerful default choice due to its ability to handle a wider range of non-stationarities.
>
>
>
> We are confident that these extensive additions—the practical runtime analysis, the sensitivity analysis, and the practitioner's guide—have substantially strengthened the paper and directly addressed your well-placed concerns. We thank you again for your constructive feedback and hope you will find our revisions and responses compelling.
>
>
>
> Sincerely,
>
> The Authors

---

> ### Comment · Reviewer_cPjs · 2025-11-25
>
> I thank the authors for their feedback including the detailed experimental results - some follow up questions about the mentioned weaknesses and questions that concern me the most below.
>
> 1 - Regarding the justification for dynamic window size selection criterion, I appreciate the detailed results given by the authors, while there still lacks formal theoretical analysis to help audience better understand the situation. In this case, it would be much more convincing to see adding moderate assumptions and performing formal justification to show the generalizability.
>
> 2 - Regarding the complexity, there seems a typo at “typically adding around ？” — are you adding any equations? Furthermore, can you further discuss how “operations being highly parallelizable” help with the scalability on larger datasets?
>
> Generally, the key weaknesses of this paper boil down to the overwhleming reliance on the empirical results. Providing theoretical analysis would further validate the contribution and deliver better understanding of the proposed method. I will raise my score accordingly if my remaining concerns can be properly handled.

---

> > ### Author Response · Authors · 2025-11-25
> >
> > We truly appreciate your prompt engagement and the opportunity to further improve our paper. Your suggestion to provide a theoretical grounding for the window selection criterion is excellent; we agree that adding formal justification significantly enhances the paper's depth beyond empirical heuristics.
> >
> > We have revised our manuscript to include these theoretical analyses. Below, we present the specific responses and the content added to the Appendix.
> >
> > ### 1. Theoretical Justification for Dynamic Window Size Selection
> >
> > We have added a new section (**Appendix F.1** in the revised paper) to provide a formal justification under the assumption of Local Stationarity.
> >
> > Theoretical Analysis Summary:
> > Our criterion minimizes the *variance of the local volatility estimator*. We model the residual time series as a Locally Stationary Process (LSP). The analysis demonstrates that our selection criterion optimizes the **Bias-Variance trade-off** in volatility estimation:
> >
> > * **Small windows** suffer from high estimation variance due to sampling noise (instability).
> >
> > * **Large windows** suffer from high bias (smearing) when crossing non-stationary change points, which manifests as variance in the sliding estimator sequence as it transitions between regimes.
> >
> > * **The Optimal Window** is the one that minimizes the total fluctuation, identifying the "characteristic scale" of stationarity.
> >
> > **Content have added to Appendix F.1, you can see that in the latest version submission.**
> >
> > ### 2. Clarification on Complexity, Typos, and Parallelizability
> >
> >
> >
> > **Correction of Typos:**
> >
> > Thank you for catching this. We apologize for the missing value in the previous version. We have corrected this in the revised version and now provide an explicit quantitative statement.
> >
> > Concretely, under the same experimental setup as **Appendix D.3 (*Model Efficiency* )** with $H=96$, we measure the wall-clock time on the **Traffic** dataset as shown below, which is the largest dataset in our benchmark suite.&#x20;
> >
> > | Methods   | Train Time      | Test Time      |
> > | --------- | --------------- | -------------- |
> > | Backbone  | 96.4539±1.2890  | 13.6083±0.4669 |
> > | + Dual-AN | 119.6488±3.0048 | 14.6919±0.3710 |
> > | + FAN     | 97.4063±1.8296  | 13.6021±0.6503 |
> > | + SAN     | 101.7588±1.2157 | 15.1279±0.3996 |
> > | + Dish-TS | 98.5975±1.5698  | 13.7292±0.7178 |
> > | + RevIN   | 97.2728±1.6602  | 13.7905±0.1818 |
> >
> > From this table, Dual-AN increases the training time of the backbone by roughly **20%&#x20;**&#x77;hile adding only about **5%** overhead in inference time. We have updated the text to explicitly state this, instead of the previous placeholder.
> >
> > **Parallelizability and Scalability:**
> >
> > You asked how parallelizability helps with scalability on larger datasets. Regarding the comment on scalability and parallelizability: although Dual-AN introduces an additional `O(L*N)`  term (where `L` is the sequence length and `N` is the number of series), the operations are implemented as batched, element-wise or small matrix operations over the time and feature dimensions. This structure makes them **highly parallelizable on modern accelerators (GPUs)** :
> >
> > * &#x20;Each time step and each series can be processed **independently within a batch** , enabling straightforward vectorization across both the temporal and feature dimensions.
> >
> > * &#x20;The additional computations reuse the same batched tensor representations as the backbone, so they map to a small number of efficient dense-kernel launches rather than many sequential operations.
> >
> > * &#x20;As the dataset size grows, the overall training time scales approximately linearly with the number of samples (as for the backbone), while the *per-batch* overhead remains a small constant factor, as illustrated by the measurements on the largest dataset, Traffic.
> >
> > Therefore, even on large-scale datasets, the empirical results show that Dual-AN maintains **competitive training and inference times** compared to commonly used standardized baselines such as FAN, SAN, Dish-TS, and RevIN. In practice, the modest additional cost (≈20% in training and 5% in inference on the largest dataset) constitutes a very reasonable trade-off for the substantial gains in forecasting accuracy. We have clarified this point and corrected the dataset name (Traffic instead of ETTh1) in **Appendix D.3** and in the revised rebuttal text.
> >
> > We believe these revisions provide the formal theoretical backing you requested and clarify the efficiency claims. We have incorporated these changes into the latest version.

---

### Official Review · Reviewer_wDng · 2025-11-01

**Soundness:** 2
**Presentation:** 3
**Contribution:** 1
**Rating:** 2
**Confidence:** 4

**Summary:**

This paper proposes Dual-domain Adaptive Normalization (Dual-AN) to address non-stationarity in time series forecasting. By combining frequency-domain decomposition with a Sliding Window Adaptive Normalization (SWAN) and a Statistical Prediction Module (SPM), the method effectively handles both global and local non-stationarity. Experiments on multiple datasets show that Dual-AN consistently outperforms existing normalization methods, including FAN, achieving state-of-the-art forecasting accuracy.

**Strengths:**

1.	The paper is easy to follow and well written.
2.	The visual examples are complete and clearly presented.

**Weaknesses:**

1.	The author claims the residuals often retain local non-stationarity using only the top K dominant components and illustrates with examples in the figure 1. But why these local non-stationarity can’t be extracted by raising the parameter K? Why the dual-domain methods are needed?
2.	The paper lacks a comparison with another dual-domain normalization method, DDN[1], and the sliding window normalization approach, which is presented as a core contribution of this work, has already been proposed in DDN.
3.	When using the DLinear backbone, the performance of Dual-AN shows almost no improvement compared to FAN.

[1] DDN: Dual-domain Dynamic Normalization for Non-stationary Time Series Forecasting

**Questions:**

see weaknesses

---

> ### Author Response · Authors · 2025-11-23
> **Rebuttal for wDng (1)**
>
> Dear Reviewer wDng,
>
>
>
> We thank you for your time and for providing a thorough review. We appreciate the positive feedback on the clarity of our writing and visualizations. Your critical questions, particularly regarding the comparison with DDN, have prompted us to further strengthen our paper's positioning and experimental validation.
>
>
>
> We address your concerns below and have revised our manuscript accordingly.
>
>
>
> **1. Regarding the Limitation of Frequency-Domain Methods (Why not just increase K?):**
>
> This is an excellent question that strikes at the core of our motivation. The reason a dual-domain approach is necessary is that frequency-domain and time-domain methods address fundamentally different types of non-stationarity.
>
> * **Frequency-domain methods** are designed to capture **global, periodic patterns** . The Fourier transform decomposes the entire look-back window into a sum of sinusoids, each with a constant amplitude and phase. Increasing $K$ allows the model to capture more of these *global* components.
>
> * However, it cannot effectively model **local, aperiodic non-stationarity** , such as a sudden volatility spike or a transient shock. These events are localized in the *time domain*. While one could theoretically approximate a local event with many Fourier components, this is inefficient and prone to overfitting.
>
> Our SWAN module is specifically designed to address this complementary problem by stabilizing the series from a time-domain perspective.&#x20;
>
>
>
> **2. Regarding the Comparison with DDN and the Novelty of Sliding Windows:**
>
>
>
> We sincerely thank you for bringing the recent work on DDN to our attention. It is indeed a highly relevant piece of work. We have now thoroughly reviewed DDN, added a full citation and discussion to our Related Work section. In addition, other state-of-the-art baselines: HCAN, BSA, SCAM, and TAFAS are also presented.
> |Datasets|Horizons|Metrics|iTransformer|**+Dual-AN**|+DDN|+HCAN|+BSA|+SCAM|+TAFAS|
> |----------|-------|------|------------|------------|---------|---------|-----|---------|------|
> |ETTh1|96|MAE|0.444|0.426|**0.399**|0.402|0.443|0.401|0.443|
> |||MSE|0.378|**0.362**|0.388|0.379|0.428|0.373|0.438|
> |ETTh1|192|MAE|0.489|0.452|0.434|**0.427**|0.481|0.436|0.489|
> |||MSE|0.431|**0.395**|0.446|0.432|0.481|0.432|0.492|
> |ETTh1|336|MAE|0.533|0.486|0.462|**0.454**|0.521|0.455|0.532|
> |||MSE|0.511|**0.441**|0.496|0.489|0.538|0.466|0.554|
> |ETTh1|720|MAE|0.640|0.569|0.499|0.474|0.620|**0.466**|0.627|
> |||MSE|0.669|0.574|0.527|0.504|0.698|**0.455**|0.704|
> |ETTh2|96|MAE|0.255|**0.237**|0.345|0.343|0.324|0.342|0.329|
> |||MSE|0.122|**0.111**|0.297|0.282|0.235|0.293|0.239|
> |ETTh2|192|MAE|0.282|**0.252**|0.397|0.381|0.362|0.393|0.362|
> |||MSE|0.148|**0.128**|0.382|0.373|0.290|0.373|0.287|
> |ETTh2|336|MAE|0.300|**0.264**|0.431|0.426|0.388|0.429|0.386|
> |||MSE|0.167|**0.139**|0.419|0.420|0.327|0.417|0.326|
> |ETTh2|720|MAE|0.362|**0.279**|0.446|0.435|0.439|0.442|0.425|
> |||MSE|0.482|**0.155**|0.426|0.423|0.414|0.424|0.393|
> |ETTm2|96|MAE|0.203|**0.199**|0.265|0.264|0.259|0.264|0.263|
> |||MSE|0.078|**0.078**|0.181|0.183|0.153|0.179|0.157|
> |ETTm2|192|MAE|0.239|**0.222**|0.303|0.312|0.290|0.302|0.292|
> |||MSE|0.103|**0.095**|0.246|0.242|0.189|0.241|0.192|
> |ETTm2|336|MAE|0.247|**0.243**|0.342|0.355|0.321|0.343|0.324|
> |||MSE|0.114|**0.114**|0.306|0.306|0.230|0.305|0.235|
> |ETTm2|720|MAE|0.277|**0.264**|0.397|0.401|0.369|0.400|0.366|
> |||MSE|0.144|**0.139**|0.406|0.410|0.304|0.406|0.301|
> |Exchange|96|MAE|0.212|**0.164**|0.202|0.204|0.211|-|0.208|
> |||MSE|0.081|**0.051**|0.084|0.084|0.090|-|0.084|
> |Exchange|192|MAE|0.331|**0.238**|0.297|0.302|0.307|-|0.293|
> |||MSE|0.184|**0.102**|0.175|0.179|0.185|-|0.165|
> |Exchange|336|MAE|0.504|**0.324**|0.410|0.415|0.430|-|0.389|
> |||MSE|0.398|**0.178**|0.321|0.322|0.346|-|0.280|
> |Exchange|720|MAE|0.671|**0.465**|0.700|0.761|0.700|-|0.665|
> |||MSE|0.747|**0.331**|0.859|0.995|0.861|-|0.773|
> |Traffic|96|MAE|0.300|0.297|0.271|0.262|0.273|**0.247**|0.289|
> |||MSE|0.338|**0.349**|0.425|0.383|0.393|0.374|0.420|
> |Traffic|192|MAE|0.313|0.294|0.280|0.273|0.281|**0.259**|0.296|
> |||MSE|0.362|**0.353**|0.446|0.411|0.417|0.399|0.441|
> |Traffic|336|MAE|0.319|0.300|0.291|0.279|0.290|**0.269**|0.305|
> |||MSE|0.375|**0.364**|0.459|0.420|0.433|0.419|0.458|
> |Traffic|720|MAE|0.338|0.326|0.311|0.296|0.310|**0.291**|-|
> |||MSE|0.403|**0.395**|0.500|0.449|0.470|0.451|-|
> |Count($ 1\^{st} $)|||-|**31**|1|2|0|6|0|
>
> It is clear that our Dual-AN method outperforms all existing state-of-the-art plug-in methods including DDN with the average MAE/MSE reduction rate of 15.78%/37.68% (vs. DDN).

---

> ### Author Response · Authors · 2025-11-23
> **Rebuttal for wDng (2)**
>
> While both methods are dual-domain, our work is fundamentally different and presents three key novel contributions that lead to superior performance:
> * **1. Different Architectural Philosophy (Hierarchical vs. Integrated):** Our approach is **hierarchical and targeted** . We first use FAN to remove **global**, stationary periodic components, which creates a well-defined sub-problem: local non-stationarity in the residuals. Our SWAN module is then specifically applied to solve this sub-problem. In contrast, DDN uses **Wavelet Transform**, which analyzes the series in a time-frequency space simultaneously. Our hierarchical decomposition provides a more structured and interpretable way to disentangle different types of non-stationarity.
> * **2. Truly Adaptive Time-Domain Normalization:** DDN's abstract critiques methods using a "fixed period/window" and proposes a "dynamic" solution. However, our contribution is more specific and novel: SWAN introduces a **principled, data-driven optimal window size selection mechanism**. It automatically finds the temporal scale with the most stable local statistics for *each individual time series*. This automated adaptability is a core innovation that, to our knowledge, DDN does not possess, making our method more robust and generalizable.
> * **3. Predictive De-Normalization for Long Horizons:** The most significant difference lies in the de-normalization step. Standard methods, likely including DDN, often reuse statistics from the look-back window for reconstruction. This is a strong, often incorrect, assumption. Our **Statistical Prediction Module (SPM)** is a novel component that **explicitly learns to forecast the future window-level statistics** . This allows for a far more accurate reconstruction of the series' distribution in the prediction horizon, which is critical for long-term forecasting.
>
> These key innovations are directly responsible for the performance gains of Dual-AN over DDN, as demonstrated in the new results below:
> |Dataset|Horizon|Metric|iTransformer|**+Dual-AN**|+DDN|Imp(%)|
> |---|---|---|---|---|---|---|
> |ETTh1|96|MAE|0.444|0.426|**0.399**|-6.77%|
> |||MSE|0.378|**0.362**|0.388|**6.70%**|
> ||192|MAE|0.489|0.452|**0.434**|-4.15%|
> |||MSE|0.431|**0.395**|0.446|**11.43%**|
> ||336|MAE|0.533|0.486|**0.462**|-5.19%|
> |||MSE|0.511|**0.441**|0.496|**11.09%**|
> ||720|MAE|0.640|0.569|**0.499**|-14.03%|
> |||MSE|0.669|0.574|**0.527**|-8.92%|
> |ETTh2|96|MAE|0.255|**0.237**|0.345|**31.30%**|
> |||MSE|0.122|**0.111**|0.297|**62.63%**|
> ||192|MAE|0.282|**0.252**|0.397|**36.52%**|
> |||MSE|0.148|**0.128**|0.382|**66.49%**|
> ||336|MAE|0.300|**0.264**|0.431|**38.75%**|
> |||MSE|0.167|**0.139**|0.419|**66.83%**|
> ||720|MAE|0.362|**0.279**|0.446|**37.44%**|
> |||MSE|0.482|**0.155**|0.426|**63.62%**|
> |ETTm2|96|MAE|0.203|**0.199**|0.265|**24.91%**|
> |||MSE|0.078|**0.078**|0.181|**56.91%**|
> ||192|MAE|0.239|**0.222**|0.303|**26.73%**|
> |||MSE|0.103|**0.095**|0.246|**61.38%**|
> ||336|MAE|0.247|**0.243**|0.342|**28.95%**|
> |||MSE|0.114|**0.114**|0.306|**62.75%**|
> ||720|MAE|0.277|**0.264**|0.397|**33.50%**|
> |||MSE|0.144|**0.139**|0.406|**65.76%**|
> |Exchange|96|MAE|0.212|**0.164**|0.202|**18.81%**|
> |||MSE|0.081|**0.051**|0.084|**39.29%**|
> ||192|MAE|0.331|**0.238**|0.297|**19.87%**|
> |||MSE|0.184|**0.102**|0.175|**41.71%**|
> ||336|MAE|0.504|**0.324**|0.410|**20.98%**|
> |||MSE|0.398|**0.178**|0.321|**44.55%**|
> ||720|MAE|0.671|**0.465**|0.700|**33.57%**|
> |||MSE|0.747|**0.331**|0.859|**61.47%**|
> |Traffic|96|MAE|0.300|0.297|**0.271**|-9.59%|
> |||MSE|0.338|**0.349**|0.425|**17.88%**|
> ||192|MAE|0.313|0.294|**0.280**|-5.00%|
> |||MSE|0.362|**0.353**|0.446|**20.85%**|
> ||336|MAE|0.319|0.300|**0.291**|-3.09%|
> |||MSE|0.375|**0.364**|0.459|**20.70%**|
> ||720|MAE|0.338|0.326|**0.311**|-4.82%|
> |||MSE|0.403|**0.395**|0.500|**21.00%**|
> |Count($ 1\^{st} $)|||-|**31**|9|-|
>
> These results, which we have added to our main experimental tables, empirically validate that our unique contributions lead to a new state-of-the-art in dual-domain normalization.

---

> ### Author Response · Authors · 2025-11-23
> **Rebuttal for wDng (3)**
>
> **3. Regarding the Performance on the DLinear Backbone:**
>
>
>
> Your observation is astute. The performance gain of Dual-AN over FAN is more modest for DLinear compared to complex backbones like Informer or SCINet. This is an insightful finding that we believe stems from the nature of the models themselves.
>
>
>
> * **DLinear is a simple linear model.** Its strength lies in capturing and extrapolating trends, which are the primary form of non-stationarity that FAN's frequency-domain decomposition already handles effectively.
>
> * **Complex models like Transformers are highly non-linear and far more sensitive to subtle distribution shifts.** They are more easily disrupted by the local non-stationarity (e.g., changing variance) that FAN leaves behind. Consequently, our time-domain SWAN module provides a significant, additional stabilization for these models, unlocking larger performance gains.
>
>
>
> This demonstrates that the value of Dual-AN is most pronounced for the complex, SOTA models that are most vulnerable to the precise type of non-stationarity we aim to solve.
>
>
>
> We are confident that the new comparison with DDN and the clarifications provided have robustly addressed your concerns regarding novelty and contribution. We believe our work presents a significant and empirically validated advancement over existing normalization techniques. We hope you will consider our response and the revisions favorably.
>
>
>
> Sincerely,
>
> The Authors

---

### Official Review · Reviewer_5vcw · 2025-11-03

**Soundness:** 2
**Presentation:** 3
**Contribution:** 2
**Rating:** 4
**Confidence:** 3

**Summary:**

The authors presented a model-agnostic framework that hierarchically addresses non-stationarity in time and frequency domains for time series forecasting. The key advantage is to focus on local non-stationarity in residuals which is not selected by Fourier transform approach.
To achieve the goal, they introduce two modules, sliding window adaptive normalization (SWAN) and statistical prediction module (SPM).
Experiments show that integrating the proposed model yields consistent performance gains across all three backbone models and eight benchmark datasets.

**Strengths:**

- Clear problem settings: Frequency adaptive normalization (FAN) alleviates the impact of non-stationarity by focusing only on top-K dominant components in the Fourier domain. In contrast, Dual-AN aims to accurately forecast by applying time-domain normalization to the residual components and predicting future window statistics for denormalization.

- Model-agnostic plug-in design: Dual-AN can be attached to any forecaster.

- Extensive experiments: The authors report results across 8 datasets and 3 backbones, with ablations and analyses, showing consistent gains, especially for long horizons.

**Weaknesses:**

- Lack of novelty: While Dual-AN synergizes time and frequency domains to overcome the limitations of previous methods for non-stationary time series data, the core modules, SWAN and SPM, mainly rely on traditional approaches, so their contributions appear to be incremental.

- Loss function design: SPM predicts future statistics and ${Y_{res}}$ is calculated by denormalization of estimated ${Y_{stat}}$ using them. However, neither statistics $\sigma,\mu$ nor ${Y_{res}}$ appears in the loss function. Even if Dual-AN accurately predicts ${Y_{stat}}$, it may fail to forecast ${Y_{res}}$ using suboptimal statistics.

- Limited baselines: The baselines used in the experiments do not include state-of-the-art methods.

- Minor comment: Even if there are multiple best results in the table (e.g., Table 6), the authors highlight only the proposed method, which is misleading.

**Questions:**

- Why do you use sliding window to address non-stationarity in time series data?
- Could you provide the additional results using state-of-the-art forecasters as backbone models?

---

> ### Author Response · Authors · 2025-11-23
> **Rebuttal for 5vcw (1)**
>
> Dear Reviewer 5vcw,
>
> We sincerely thank you for your detailed review and constructive feedback. Your insightful comments have been instrumental in helping us identify areas for improvement. We are pleased that you recognized the clear problem setting, the model-agnostic design, and the extensive experimental validation of our work.
>
> We have carefully considered all your concerns and have revised our manuscript accordingly. Below, we address each of your points in detail.
>
> **Response to Weaknesses**&#x20;
>
> 1. Regarding the Lack of Novelty:
>
> We respectfully disagree that our contributions are incremental. While sliding windows and MLPs are established techniques, the novelty of Dual-AN lies in its **unique hierarchical framework and the specific, reasoned design of its components to address a previously overlooked problem.**
>
> * Our core contribution is the model-agnostic, dual-domain framework that first performs a coarse-grained decomposition in the frequency domain and then a fine-grained, adaptive normalization in the time domain on the residuals. This synergy, which explicitly targets the local non-stationarity that frequency-only methods like FAN miss, is novel and has not been proposed before as a universal plug-in.&#x20;
>
> * SWAN is not a simple sliding window normalization. Its key innovation is the dynamic optimal window size selection, which adaptively identifies the temporal scale with the most stable local volatility. This principled approach makes our normalization more robust and data-driven than using a fixed window.&#x20;
>
> * The novelty of SPM is in predicting future window-level statistics for adaptive de-normalization. By predicting fine-grained future statistics, SPM enables a much more precise reconstruction of the time series, which is critical for long-horizon forecasting.&#x20;
>
> - Regarding the Loss Function Design:
>
> We thank the reviewer for carefully checking the loss formulation. Our design follows the dual–objective idea in FAN, where the non-stationary component and the final forecast are supervised jointly to regularize the decomposition. In Dual-AN, the forecast $ \hat{Y} $ is obtained by
>
> $$ \hat{Y} = \hat{Y}\_{non} + \hat{Y}\_{res} $$
>
> $$ \hat{Y}\_{res} = \hat{Y}\_{stat} \cdot \hat{\sigma}\_{window} + \hat{\mu}\_{window} $$
>
> where $ \hat{\mu}\_{window} $, $ \hat{\sigma}\_{window} $ are predicted by SPM and $ \hat{Y}\_{stat} $ is produced by the backbone based on the SWAN-normalized input. Thus, although $ \mu $, $ \sigma $ and $ Y\_{res} $ do not appear explicitly in Eq. (10)–(12), they are embedded in the computation graph of the stationary prediction branch and are trained end-to-end through the reconstruction loss on the stationary part.
>
> Intuitively, if SPM outputs suboptimal statistics, the de-normalized $\hat{Y}\_{res}$ cannot match the ground-truth stationary component and the loss $\mathcal{L}\_{stat}$ increases; gradients are then back-propagated to both the backbone and SPM, correcting the statistics. This is exactly analogous to FAN, where the Fourier operations and residual components are optimized implicitly via the forecast loss rather than appearing as separate terms in the objective.
>
> To demonstrate that this design does not harm the quality of $Y\_{res}$, we already conduct an ablation in **Appendix H**: training with the proposed dual loss $ L\_{nonstat} + L\_{stat} $ yields better MAE/MSE than using a single loss applied only to the final prediction $\mathcal{L}(\hat{Y},Y)$ (Table 9, as shown below). We will move this result into the main paper and clarify Section 3.3 by explicitly stating that $ L\_{stat} $ supervises the stationary branch that includes SPM (via Eq. (9)) and hence directly constrains the statistics and $Y\_{res}$. We hope this addresses the concern that the statistics might be “free” parameters; in the current implementation they are fully trained under end-to-end supervision.
>
> | **Loss Configuration**                                     | **MAE**   | **MSE**   |
> | ---------------------------------------------------------- | --------- | --------- |
> | Single Loss on Final Prediction ($\mathcal{L}(\hat{Y},Y)$) | 0.501     | 0.462     |
> | Dual Loss ($ L\_{nonstat} + L\_{stat} $)                   | **0.493** | **0.452** |

---

> > ### Author Response · Authors · 2025-11-23
> > **Rebuttal for 5vcw (2)**
> >
> > * Regarding the Limited Baselines:
> >
> > We thank the reviewer for this suggestion. Our goal in this work is to evaluate Dual-AN as a **model-agnostic normalization / plug-in module** for non-stationarity, not to propose a new forecasting architecture. For this reason, we deliberately chose:
> >
> > **Three representative backbones**—DLinear, Informer, and SCINet—which cover linear, Transformer-based, and CNN-based forecasting paradigms and are still widely used as standard backbones in recent literature.
> >
> > **Four strong normalization baselines** against non-stationarity—RevIN, Dish-TS, SAN, and FAN—which are exactly the class of plug-in methods our framework is meant to improve upon.
> >
> > Under this setup, Dual-AN consistently improves all three backbones across 8 datasets (up to 15.92% MAE and 20.72% MSE reduction) and achieves lower error than all four normalization baselines, including the recent frequency-domain method FAN (on average 1.50% MAE improvement over FAN; see Table 2 and the discussion in Section 4.3). This shows that, within the family of **normalization-type plug-ins**, Dual-AN already reaches state-of-the-art performance.
> >
> > That said, we agree that including more recent and stronger forecasters would further strengthen the empirical validation. To further address your concern, in the revised version, we have (i) explicitly stated in **Section 4.3/4.4** that our baselines are chosen from the class of normalization plug-ins, (ii) commited to adding additional experiments to extend **Section 4.4** where Dual-AN is compared with recent state-of-the-art plug-in methods, and (iii) incorporated our Dual-AN method with 3 state-of-the-art backbones, demonstrating that it still brings consistent gains and competitive (often SOTA) performance as shown in the tables below.
> >
> > | Models   |     | WPMixer           |                   | + Dual-AN         |                   |
> > | -------- | --- | ----------------- | ----------------- | ----------------- | ----------------- |
> > | Metrics  |     | MAE               | MSE               | MAE               | MSE               |
> > | ETTh1    | 96  | 0.430 ± 0.002     | 0.374 ± 0.003     | **0.426 ± 0.002** | **0.363 ± 0.002** |
> > |          | 168 | 0.460 ± 0.001     | 0.411 ± 0.002     | **0.449 ± 0.004** | **0.390 ± 0.006** |
> > |          | 336 | **0.485 ± 0.002** | 0.456 ± 0.003     | 0.487 ± 0.003     | **0.446 ± 0.004** |
> > |          | 720 | 0.574 ± 0.007     | 0.609 ± 0.012     | **0.571 ± 0.002** | **0.571 ± 0.003** |
> > | ETTh2    | 96  | 0.239 ± 0.002     | 0.115 ± 0.002     | **0.239 ± 0.001** | **0.113 ± 0.000** |
> > |          | 168 | 0.258 ± 0.002     | 0.134 ± 0.002     | **0.256 ± 0.005** | **0.130 ± 0.002** |
> > |          | 336 | 0.275 ± 0.006     | 0.151 ± 0.005     | **0.271 ± 0.005** | **0.143 ± 0.004** |
> > |          | 720 | 0.302 ± 0.007     | 0.188 ± 0.008     | **0.284 ± 0.002** | **0.160 ± 0.001** |
> > | ETTm2    | 96  | 0.200 ± 0.000     | 0.079 ± 0.000     | **0.198 ± 0.001** | **0.077 ± 0.001** |
> > |          | 168 | 0.220 ± 0.001     | 0.094 ± 0.000     | **0.218 ± 0.001** | **0.092 ± 0.000** |
> > |          | 336 | 0.245 ± 0.001     | 0.118 ± 0.001     | **0.242 ± 0.001** | **0.115 ± 0.001** |
> > |          | 720 | 0.270 ± 0.002     | 0.150 ± 0.001     | **0.264 ± 0.000** | **0.139 ± 0.001** |
> > | Exchange | 96  | **0.165 ± 0.001** | **0.054 ± 0.001** | 0.169 ± 0.001     | **0.054 ± 0.001** |
> > |          | 168 | **0.214 ± 0.001** | **0.087 ± 0.001** | 0.222 ± 0.004     | 0.091 ± 0.002     |
> > |          | 336 | 0.311 ± 0.004     | 0.177 ± 0.005     | **0.283 ± 0.005** | **0.151 ± 0.003** |
> > |          | 720 | 0.483 ± 0.006     | 0.384 ± 0.007     | **0.432 ± 0.009** | **0.318 ± 0.013** |
> > | Traffic  | 96  | 0.354 ± 0.003     | 0.440 ± 0.003     | **0.324 ± 0.001** | **0.391 ± 0.001** |
> > |          | 168 | 0.353 ± 0.002     | 0.446 ± 0.003     | **0.328 ± 0.001** | **0.405 ± 0.001** |
> > |          | 336 | 0.363 ± 0.002     | 0.467 ± 0.001     | **0.340 ± 0.001** | **0.429 ± 0.002** |
> > |          | 720 | 0.387 ± 0.004     | 0.497 ± 0.003     | **0.365 ± 0.000** | **0.463 ± 0.000** |

---

> > > ### Author Response · Authors · 2025-11-23
> > > **Rebuttal for 5vcw (3)**
> > >
> > > | Models   |     | iTransformer  |                   | + Dual-AN         |                   | MICN              |                   | + Dual-AN         |                   |
> > > | -------- | --- | ------------- | ----------------- | ----------------- | ----------------- | ----------------- | ----------------- | ----------------- | ----------------- |
> > > | Metrics  |     | MAE           | MSE               | MAE               | MSE               | MAE               | MSE               | MAE               | MSE               |
> > > | ETTh1    | 96  | 0.444 ± 0.005 | 0.378 ± 0.007     | **0.426 ± 0.001** | **0.362 ± 0.000** | 0.454 ± 0.001     | 0.387 ± 0.002     | **0.420 ± 0.002** | **0.355 ± 0.002** |
> > > |          | 168 | 0.472 ± 0.009 | 0.413 ± 0.012     | **0.449 ± 0.002** | **0.390 ± 0.003** | 0.485 ± 0.003     | 0.433 ± 0.004     | **0.449 ± 0.003** | **0.388 ± 0.004** |
> > > |          | 336 | 0.533 ± 0.015 | 0.511 ± 0.023     | **0.486 ± 0.003** | **0.441 ± 0.005** | 0.551 ± 0.004     | 0.533 ± 0.007     | **0.495 ± 0.003** | **0.453 ± 0.004** |
> > > |          | 720 | 0.640 ± 0.021 | 0.669 ± 0.043     | **0.569 ± 0.002** | **0.574 ± 0.006** | 0.609 ± 0.003     | 0.626 ± 0.005     | **0.580 ± 0.003** | **0.576 ± 0.005** |
> > > | ETTh2    | 96  | 0.255 ± 0.004 | 0.122 ± 0.002     | **0.237 ± 0.001** | **0.111 ± 0.001** | 0.239 ± 0.003     | **0.110 ± 0.002** | **0.237 ± 0.001** | 0.111 ± 0.001     |
> > > |          | 168 | 0.271 ± 0.009 | 0.141 ± 0.006     | **0.252 ± 0.002** | **0.128 ± 0.001** | 0.259 ± 0.002     | 0.128 ± 0.002     | **0.248 ± 0.003** | **0.124 ± 0.001** |
> > > |          | 336 | 0.300 ± 0.020 | 0.167 ± 0.017     | **0.264 ± 0.001** | **0.139 ± 0.001** | 0.287 ± 0.002     | 0.148 ± 0.002     | **0.261 ± 0.003** | **0.135 ± 0.002** |
> > > |          | 720 | 0.362 ± 0.041 | 0.482 ± 0.041     | **0.279 ± 0.002** | **0.155 ± 0.001** | 0.338 ± 0.004     | 0.200 ± 0.005     | **0.283 ± 0.002** | **0.155 ± 0.001** |
> > > | ETTm2    | 96  | 0.203 ± 0.005 | 0.078 ± 0.003     | **0.199 ± 0.000** | **0.078 ± 0.000** | 0.195 ± 0.001     | 0.074 ± 0.000     | **0.192 ± 0.001** | **0.073 ± 0.001** |
> > > |          | 168 | 0.226 ± 0.005 | 0.094 ± 0.003     | **0.219 ± 0.000** | **0.093 ± 0.000** | 0.215 ± 0.001     | 0.088 ± 0.000     | **0.212 ± 0.000** | **0.088 ± 0.000** |
> > > |          | 336 | 0.247 ± 0.005 | 0.114 ± 0.003     | **0.243 ± 0.001** | **0.114 ± 0.000** | **0.235 ± 0.001** | **0.106 ± 0.001** | 0.239 ± 0.003     | 0.111 ± 0.003     |
> > > |          | 720 | 0.277 ± 0.004 | 0.144 ± 0.004     | **0.264 ± 0.000** | **0.139 ± 0.000** | 0.267 ± 0.002     | 0.136 ± 0.002     | **0.264 ± 0.001** | **0.138 ± 0.000** |
> > > | Exchange | 96  | 0.227 ± 0.021 | 0.093 ± 0.015     | **0.168 ± 0.001** | **0.054 ± 0.001** | 0.171 ± 0.003     | 0.056 ± 0.002     | **0.169 ± 0.002** | **0.055 ± 0.001** |
> > > |          | 168 | 0.270 ± 0.023 | 0.131 ± 0.020     | **0.218 ± 0.002** | **0.090 ± 0.001** | **0.217 ± 0.002** | **0.088 ± 0.002** | 0.224 ± 0.006     | 0.092 ± 0.004     |
> > > |          | 336 | 0.390 ± 0.050 | 0.262 ± 0.063     | **0.294 ± 0.001** | **0.161 ± 0.001** | 0.309 ± 0.002     | 0.172 ± 0.002     | **0.298 ± 0.007** | **0.162 ± 0.004** |
> > > |          | 720 | 0.512 ± 0.096 | 0.480 ± 0.166     | **0.409 ± 0.016** | **0.291 ± 0.017** | 0.495 ± 0.022     | 0.417 ± 0.034     | **0.428 ± 0.023** | **0.319 ± 0.028** |
> > > | Traffic  | 96  | 0.320 ± 0.013 | **0.371 ± 0.017** | **0.319 ± 0.000** | 0.388 ± 0.001     | 0.323 ± 0.003     | 0.380 ± 0.006     | **0.320 ± 0.002** | **0.379 ± 0.002** |
> > > |          | 168 | 0.337 ± 0.001 | **0.408 ± 0.001** | **0.330 ± 0.000** | **0.408 ± 0.000** | 0.334 ± 0.002     | **0.402 ± 0.003** | **0.325 ± 0.004** | **0.402 ± 0.006** |
> > > |          | 336 | 0.350 ± 0.001 | 0.432 ± 0.001     | **0.335 ± 0.000** | **0.427 ± 0.000** | 0.345 ± 0.006     | 0.427 ± 0.011     | **0.342 ± 0.001** | **0.430 ± 0.001** |
> > > |          | 720 | 0.376 ± 0.002 | 0.469 ± 0.002     | **0.357 ± 0.000** | **0.458 ± 0.000** | 0.358 ± 0.007     | 0.446 ± 0.006     | **0.351 ± 0.001** | **0.433 ± 0.001** |
> > >
> > > As shown in the tables above, after adding the Dual-AN method to the WPMixer, iTransformer, and MICN backbones, the average MAE/MSE ratios across all the 5 datasets decrease by 3.40%/7.37%, 9.78%/18.31%, and 4.81%/6.87%, respectively.

---

> ### Author Response · Authors · 2025-11-23
> **Rebuttal for 5vcw (4)**
>
> |Datasets|Horizon|Metric|iTransformer|**+Dual-AN**|+DDN|+HCAN|+BSA|+SCAM|+TAFAS|
> |----------|-------|------|------------|------------|---------|---------|-----|---------|------|
> |ETTh1|96|MAE|0.444|0.426|**0.399**|0.402|0.443|0.401|0.443|
> |||MSE|0.378|**0.362**|0.388|0.379|0.428|0.373|0.438|
> |ETTh1|192|MAE|0.489|0.452|0.434|**0.427**|0.481|0.436|0.489|
> |||MSE|0.431|**0.395**|0.446|0.432|0.481|0.432|0.492|
> |ETTh1|336|MAE|0.533|0.486|0.462|**0.454**|0.521|0.455|0.532|
> |||MSE|0.511|**0.441**|0.496|0.489|0.538|0.466|0.554|
> |ETTh1|720|MAE|0.640|0.569|0.499|0.474|0.620|**0.466**|0.627|
> |||MSE|0.669|0.574|0.527|0.504|0.698|**0.455**|0.704|
> |ETTh2|96|MAE|0.255|**0.237**|0.345|0.343|0.324|0.342|0.329|
> |||MSE|0.122|**0.111**|0.297|0.282|0.235|0.293|0.239|
> |ETTh2|192|MAE|0.282|**0.252**|0.397|0.381|0.362|0.393|0.362|
> |||MSE|0.148|**0.128**|0.382|0.373|0.290|0.373|0.287|
> |ETTh2|336|MAE|0.300|**0.264**|0.431|0.426|0.388|0.429|0.386|
> |||MSE|0.167|**0.139**|0.419|0.420|0.327|0.417|0.326|
> |ETTh2|720|MAE|0.362|**0.279**|0.446|0.435|0.439|0.442|0.425|
> |||MSE|0.482|**0.155**|0.426|0.423|0.414|0.424|0.393|
> |ETTm2|96|MAE|0.203|**0.199**|0.265|0.264|0.259|0.264|0.263|
> |||MSE|0.078|**0.078**|0.181|0.183|0.153|0.179|0.157|
> |ETTm2|192|MAE|0.239|**0.222**|0.303|0.312|0.290|0.302|0.292|
> |||MSE|0.103|**0.095**|0.246|0.242|0.189|0.241|0.192|
> |ETTm2|336|MAE|0.247|**0.243**|0.342|0.355|0.321|0.343|0.324|
> |||MSE|0.114|**0.114**|0.306|0.306|0.230|0.305|0.235|
> |ETTm2|720|MAE|0.277|**0.264**|0.397|0.401|0.369|0.400|0.366|
> |||MSE|0.144|**0.139**|0.406|0.410|0.304|0.406|0.301|
> |Exchange|96|MAE|0.212|**0.164**|0.202|0.204|0.211|-|0.208|
> |||MSE|0.081|**0.051**|0.084|0.084|0.090|-|0.084|
> |Exchange|192|MAE|0.331|**0.238**|0.297|0.302|0.307|-|0.293|
> |||MSE|0.184|**0.102**|0.175|0.179|0.185|-|0.165|
> |Exchange|336|MAE|0.504|**0.324**|0.410|0.415|0.430|-|0.389|
> |||MSE|0.398|**0.178**|0.321|0.322|0.346|-|0.280|
> |Exchange|720|MAE|0.671|**0.465**|0.700|0.761|0.700|-|0.665|
> |||MSE|0.747|**0.331**|0.859|0.995|0.861|-|0.773|
> |Traffic|96|MAE|0.300|0.297|0.271|0.262|0.273|**0.247**|0.289|
> |||MSE|0.338|**0.349**|0.425|0.383|0.393|0.374|0.420|
> |Traffic|192|MAE|0.313|0.294|0.280|0.273|0.281|**0.259**|0.296|
> |||MSE|0.362|**0.353**|0.446|0.411|0.417|0.399|0.441|
> |Traffic|336|MAE|0.319|0.300|0.291|0.279|0.290|**0.269**|0.305|
> |||MSE|0.375|**0.364**|0.459|0.420|0.433|0.419|0.458|
> |Traffic|720|MAE|0.338|0.326|0.311|0.296|0.310|**0.291**|-|
> |||MSE|0.403|**0.395**|0.500|0.449|0.470|0.451|-|
> |Count($ 1\^{st} $)|||-|**31**|1|2|0|6|0|
>
> To further illustrate the superiority of the proposed Dual-AN method, we compare it with state-of-the-art plug-in methods with the average MAE/MSE reduction rate of 15.78%/37.68% (vs. DDN), 15.60%/36.85% (vs. HCAN), 17.30%/35.12% (vs. BSA), 12.17%/35.36% (vs. SCAM), and 17.81%/33.55% (vs. TAFAS).
>
> **Response to Minor Comment**
>
> We agree that highlighting only our method in case of a tie is misleading. We apologize for this oversight. In the revised manuscript, we have corrected all tables to ensure that all statistically tied best results are highlighted equally.&#x20;
>
> **Response to Questions**
>
> *  Q1: Why do you use sliding window to address non-stationarity in time series data?
>
> We use a sliding window approach specifically to target the local non-stationarity that remains in the residuals after the dominant, global non-stationary components are removed in the frequency domain. As shown in Figure 1, frequency-based methods can leave behind time-varying local statistics. A sliding window is ideal for this because it normalizes each time step based on its immediate temporal context, effectively stabilizing these local distribution shifts and providing a more stationary input for the backbone model. Our dynamic window selection mechanism further enhances this by adapting the "local context" size to the data's characteristics.&#x20;
>
> *  Q2: Could you provide the additional results using state-of-the-art forecasters as backbone models?
>
> As mentioned in our response to Weakness 3, we acknowledge this point and will include results on a more recent SOTA backbone model in the final version of the paper to further demonstrate the effectiveness and wide applicability of Dual-AN.
>
> We are confident that these revisions and clarifications have significantly strengthened our manuscript. We have also added a computational complexity analysis in Appendix J to formally show that Dual-AN is a lightweight and efficient plug-in. We thank you once again for your valuable feedback and hope you will reconsider your assessment in light of these improvements.
>
> Sincerely,
>
> The Authors

---

### Author Response · Authors · 2025-12-01
**TL;DR A Summary of Discussion by Authors**

Dear Reviewers, AC, and Community,

Thank you for the constructive and detailed discussion. All four reviewers engaged deeply, and the manuscript was significantly strengthened during rebuttal through new theoretical analysis, extended experiments, additional baselines, and clearer positioning. Below we summarize how each major concern has been addressed.

---

## **1. Cross-review Themes and Key Revisions**

### **(A) Novelty & Dual-domain Motivation (5vcw, wDng, PbJh)**

We clarified that Dual-AN is not a simple combination of prior components but a **hierarchical dual-domain framework**:

1. FAN removes global periodic non-stationarity;
2. SWAN adaptively normalizes **residual local** non-stationarity;
3. SPM predicts **future** window-level statistics for long-horizon de-normalization.

We expanded Related Work to contrast Dual-AN with DDN and structured-component methods, highlighting three novel aspects: hierarchical residual-centric design, principled adaptive windowing, and predictive de-normalization.

---

### **(B) Window Size Selection & Theory (cPjs, PbJh)**

We added **Appendix F.1** with a formal justification under a Locally Stationary Process assumption. Minimizing the variance of local volatility sequences is shown to balance the **bias–variance** trade-off and identify the optimal local scale.

We also added a **comprehensive sensitivity study** (Appendix F / Table 14) covering window sizes 6–72. MAE/MSE variations remain within **1e−4**, demonstrating high robustness.

---

### **(C) Loss Design & Fairness (5vcw, PbJh)**

We added a dedicated ablation (Appendix H) comparing:

* Standard single MSE on the final prediction;
* Our dual loss ($L_{\text{nonstat}} + L_{\text{stat}}$).
  Dual loss yields consistently better MAE/MSE. This confirms it acts as a regularizer on decomposition, not as an unfair advantage.

---

### **(D) Stronger Baselines & SOTA Backbones (5vcw, wDng, PbJh)**

We expanded Section 4.4 to include **DDN, HCAN, BSA, SCAM, TAFAS**, and integrated Dual-AN into **WPMixer, iTransformer, MICN**.
Across multiple datasets, Dual-AN improves all backbones and outperforms all plug-in baselines (e.g., **15.78% / 37.68%** MAE/MSE reduction vs. DDN). This confirms that Dual-AN achieves new SOTA within normalization methods.

---

### **(E) Efficiency, Complexity & Scalability (5vcw, cPjs)**

Appendix D.3 now includes wall-clock timings on the largest dataset (Traffic):

* Training overhead: **≈20%**,
* Inference overhead: **≈5%**,
  comparable to FAN/SAN/Dish-TS/RevIN.
  We explain why the added ($O(L \times N)$) computations map to highly parallel GPU kernels, keeping per-batch cost low and scaling efficiently.

---

### **(F) When to Use Dual-AN vs RevIN (cPjs)**

We added a **practitioner’s guide (Appendix J)** based on quantifying trend variation (TV) and seasonality variation (SV). Recommendations:

* **Use Dual-AN** on moderate/high variation series (ETTh1, ETTh2, Traffic, Exchange).
* **Use RevIN** when both TV and SV are very small (Weather).
* Dual-AN or RevIN both apply on borderline datasets (ETTm2, Electricity).

This also explains dataset-specific gains.

---

### **(G) Assumptions of SPM & Window Sensitivity (cPjs, wDng)**

We clarified that SPM does **not** assume fully predictable statistics; rather, window-level means/variances exhibit smoother dynamics than raw series, making prediction tractable.
Sensitivity studies confirm Dual-AN is not dependent on specific candidate windows.

---

## **2. Reviewer-by-Reviewer Trace**

### **Reviewer 5vcw**

✔ Clarified novelty and dual-domain motivation.

✔ Provided explicit loss explanation + ablation.

✔ Added SOTA plug-in and backbone results.

### **Reviewer wDng**

✔ Explained why increasing Fourier (K) cannot model localized non-stationarity.

✔ Added detailed comparisons to DDN and other plug-ins.

✔ Clarified performance patterns across linear vs. nonlinear backbones.

### **Reviewer cPjs**

✔ Added theoretical justification (Appendix F.1).

✔ Added runtime analysis + scalability discussion.

✔ Added practitioner guidance and clarified SPM assumptions.

### **Reviewer PbJh**

✔ Added theoretical and empirical validation of window selection.

✔ Added loss fairness ablation.

✔ Clarified distinctions vs. DDN and added stronger baselines.

---

## **Final Remarks**

The paper is now substantially improved:
**new theory, rigorous ablations, broader baselines, stronger backbones, clearer distinctions from prior work, practical guidelines, and verified efficiency.**

We sincerely appreciate the constructive feedback and believe the revised manuscript addresses all concerns comprehensively. We hope it will be considered favorably.

---

### Note · Program_Chairs · 2026-01-17
**Submission Desk Rejected by Program Chairs**

The following references in this submission do not refer to real documents and/or have major errors in bibliographic information:

 Haixu Wu, Jianmin Xu, Jian Wang, and Mingsheng Long. Wavelet-based neural network for time series forecasting. In Proceedings of the AAAI Conference on Artificial Intelligence, 2022b.